# Sub-second dynamics of theta-gamma coupling in hippocampal CA1

Lu Zhang[1], John Lee[2], Christopher Rozell[1,2], Annabelle C Singer[1]*

[1]Coulter Department of Biomedical Engineering, Georgia Institute of Technology and Emory University, Atlanta, United States; [2]School of Electrical and Computer Engineering, Georgia Institute of Technology, Atlanta, United States

**Abstract** Oscillatory brain activity reflects different internal brain states including neurons' excitatory state and synchrony among neurons. However, characterizing these states is complicated by the fact that different oscillations are often coupled, such as gamma oscillations nested in theta in the hippocampus, and changes in coupling are thought to reflect distinct states. Here, we describe a new method to separate single oscillatory cycles into distinct states based on frequency and phase coupling. Using this method, we identified four theta-gamma coupling states in rat hippocampal CA1. These states differed in abundance across behaviors, phase synchrony with other hippocampal subregions, and neural coding properties suggesting that these states are functionally distinct. We captured cycle-to-cycle changes in oscillatory coupling states and found frequent switching between theta-gamma states showing that the hippocampus rapidly shifts between different functional states. This method provides a new approach to investigate oscillatory brain dynamics broadly.

DOI: https://doi.org/10.7554/eLife.44320.001

## Introduction

Oscillatory brain activity is thought to play a key role in how groups of neurons interact (*Buzsáki et al., 2013*; *Colgin, 2016*; *Fries, 2015*). Furthermore, oscillatory activity serves as a read-out of internal network states: oscillations (typically recorded extracellularly) reflect rhythmic fluctuations in excitability, and therefore periods during which neurons are more or less likely to respond to excitatory inputs by generating action potentials (*Cardin, 2016*; *Fries, 2015*; *Roberts et al., 2013*; *Rohenkohl et al., 2018*; *Sohal, 2016*). As a result, oscillations are theorized to produce temporal windows for communication between neurons, sometimes called 'communication via coherence.' In fact, different frequencies of oscillations have been proposed to route communication between different subregions of the hippocampus (*Colgin, 2011*; *Colgin et al., 2009*; *Fernández-Ruiz et al., 2017*; *Lasztóczi and Klausberger, 2016*; *Schomburg et al., 2014*). Rapid changes in oscillatory activity could facilitate flexible shifts in communication between different ensembles of cells or brain regions. However, current methods to assess oscillatory activity average neural signals over long consecutive time periods (*Canolty et al., 2006*; *Le Van Quyen and Bragin, 2007*), obscuring moment-to-moment changes in these signals that may indicate rapid changes in communication between brain regions (*Huys et al., 2014*; *Tognoli and Kelso, 2009*). Thus new methods to assess dynamic changes in oscillatory activity are sorely needed. Furthermore, oscillatory activity is often characterized based only on frequency content (*Buzsáki and Draguhn, 2004*; *Wang, 2010*) even though extensive evidence points to interactions between different types of oscillations (*Axmacher et al., 2010*; *Colgin, 2011*; *Fiebelkorn et al., 2018*). Faster oscillations are often nested in slower oscillations with the faster oscillations appearing at a particular phase of the slower oscillation (*Canolty and Knight, 2010*; *Jirsa and Müller, 2013*), such as gamma (30–150 Hz) nested in theta (6–12 Hz) in the hippocampus (HPC), parietal cortex and prefrontal cortex (PFC), as well as

*For correspondence:
asinger@gatech.edu

Competing interests: The authors declare that no competing interests exist.

delta (4 Hz)-gamma (30–100 Hz) coupling in PFC and ventral tegmental area (VTA) (*Buzsáki et al., 2003*; *Fujisawa and Buzsaki, 2011*; *Scheffzük et al., 2011*; *Sirota et al., 2008*; *Tamura et al., 2017*; *Tort et al., 2013*; *Tort et al., 2009*; *Tort et al., 2008*; *Trimper et al., 2014*; *Zhang et al., 2016*). Thus methods to analyze oscillatory activity must take into account oscillatory coupling and phase relationships across frequencies in addition to the frequency content of a particular oscillation.

Oscillatory activity has been especially well-studied in HPC with close attention to the relationship between oscillations, behavior, and neural spiking activity. Three subtypes of gamma oscillations (30–150 Hz), characterized by different frequency content, nest in different phases of theta oscillations (6–12 Hz) in hippocampal CA1 (*Amemiya and Redish, 2018*; *Andersen et al., 2006*; *Buzsáki, 2006*; *Colgin et al. (2009)*; *Klausberger and Somogyi (2008)*; *Lasztóczi and Klausberger, 2016*; *Schomburg et al., 2014*). Slow gamma (30–50 Hz), which is dominant in stratum (str.) radiatum (rad), is associated with input from the CA3 subregion of HPC and is thought to play a role in memory retrieval (*Bieri et al., 2014*; *Colgin, 2015a*; *Igarashi et al., 2014*; *Tort et al., 2009*). Medium gamma (60–120 Hz), which is most active in str. lacunosum-moleculare (lm), is associated with input from layer III of the entorhinal cortex (EC3) and is thought to encode ongoing sensory information (*Bieri et al., 2014*; *Cabral et al., 2014*; *Newman et al., 2013*; *Takahashi et al., 2014*). Fast gamma (>120 Hz), is thought to represent local neural activity in str. pyramidal (pyr) of CA1 (*Schomburg et al., 2014*; *Sullivan et al., 2011*).

While prior work has examined the origins of different gamma oscillations during theta in HPC, the temporal organization of these oscillations is poorly understood because current analysis methods obscure moment-to-moment change in theta-gamma coupling. Indeed, prior work has hypothesized that CA1 rapidly shifts between inputs from CA3, which are thought important for memory retrieval, and inputs from EC, which are thought to process ongoing sensory experiences. If such rapid shifts occur, they would be reflected by rapid shifts between different types of gamma from one theta cycle to the next (*Gupta et al., 2012*; *Hasselmo et al., 2002*; *Hasselmo and Stern, 2014*; *Mizuseki et al., 2009*). Alternatively, CA1 and theta-gamma coupling may remain in a single state over multiple theta cycles.

In this paper, we describe a novel two-step analysis method to track individual theta cycles based on gamma frequency content and gamma's preferred phase of theta. First, we cluster theta-gamma coupling into different states using signal processing and machine learning methods. Second, we track moment-to-moment changes in theta-gamma coupling following Markov processes using random process theory. In the clustering phase, we found four theta-gamma coupling states without assuming the number of states that exist. These four states correspond to both previously reported theta-gamma coupling, namely slow, medium, and fast gamma (*Colgin et al., 2009*; *Schomburg et al., 2014*), and new states, namely two distinct fast gammas. In the second phase of the analysis, we tracked dynamic state changes including occurrences and dynamic transitions between states before, during, and after a spatial exploration task. We found rapid changes in theta-gamma coupling states from one theta cycle to the next. Finally, we found neural codes, specifically spatial information and phase precession differed across the identified theta-gamma states (TG states), supporting that these states have distinct functional roles. Indeed, these theta-gamma coupling states in CA1 have distinct pairwise phase consistency (PPC) with other hippocampal subregions and abundance during different behaviors and REM sleep. Together these results reveal that CA1 rapidly shifts between four theta-gamma coupling states that likely reflect distinct computational processes in the hippocampus. This new approach provides a new way to investigate and categorize oscillatory brain dynamics and their related brain states broadly without averaging across consecutive time periods.

## Results

### Community and k-means clustering separates individual theta-gamma coupling states

Previous work has observed that gamma oscillations in hippocampal CA1 differ in frequency content and preferred theta phase (*Bieri et al., 2014*; *Colgin et al., 2009*; *Lasztóczi and Klausberger, 2016*; *Schomburg et al., 2014*), and therefore we used this information to classify each theta cycle into different TG states. We calculated a frequency and theta phase power matrix (FPP, see

Materials and methods) using wavelets for each individual theta cycle from local field potentials (LFPs) recorded in hippocampal CA1 pyramidal layer (*Figure 1A*; *Figure 1—figure supplement 1*) during awake behavior. Each individual theta cycle was represented by a FPP vector with 1260 dimensions (20 phases × 81 frequencies; *Figure 1B*). Then we grouped the FPP vectors of all theta cycles for each animal using machine learning methods (*Figure 1C,D,E*, see Materials and methods). We applied k-means clustering with k determined by community clustering to categorize FPPs into 4 TG coupling states (see Materials and methods). In the data analyzed, the recording sites cover approximately the pyramidal layer (spanning 160 or 200 μm in depth depending on the number of recording sites in each shank) and do not cover all layers of the CA1 region, like str. rad or str. lm. All clusters of FPP detected across recording sites had high within-group correlations over time showing that TG state categorization was similar across channels over time (*Figure 1—figure supplement 2B* right panel). The results generated by the k-means method with four clusters, were highly robust across all recording sites within the pyramidal layer. Based on our observation, however, the peak frequency of the 'medium' gamma band (second column in *Figure 1—figure supplement 2A*, right panel) became higher (as high as high gamma) on recording sites toward the stratum oriens side of the pyramidal layer. Thus, we suggest using data recorded near the pyramidal layer center or below (toward stratum radiatum side) for this analysis method to preserve the frequency characteristics of 'medium gamma'. We also examined current source density (CSD) across recording depths for the four TG states (*Figure 1—figure supplement 3*). However because the recording sites do not span str. rad and str. lm, the input layers of CA1 and expected sources of slow and medium gamma, respectively (*Colgin et al., 2009*), the interpretation of CSD was unclear. In short, we used k-means clustering with k determined by community clustering to produce robust clustering.

## Four theta-gamma states detected during awake behaviors and REM

Four clusters were found in the LFPs and theta cycle classification was very similar across different recording depths of the pyramidal layer for each theta cycle (*Figure 1—figure supplement 2*). We then characterized these four TG states quantitatively based on their FPPs. We computed a mean FPP (m-FPP) for each cluster for each LFP recorded in the center of the pyramidal layer and identified the m-FPP's 'gamma field' (above 95% of the peak m-FPP, see Materials and methods). Using the center of gravity of the gamma field, we characterized the preferred frequency and theta-phase of the m-FPP for each TG state (*Figure 1D*, triangles). The gravity frequency and theta-phase were used as features for matching all clusters across electrodes, sessions, and animals (see Materials and methods for details).

In all rats during awake periods, we found four clusters (*Figure 1F*) that differed significantly from the other clusters in terms of preferred frequency ($p < 0.001$, $F_{3, 210}$ (gamma states)=1160.02, one-way ANOVA repeated Measures; paired t-test, $q < 0.05$, FDR correction for six comparisons) and/or theta phase (Parametric Watson-Williams multi-sample test (*Berens, 2009*); $q < 0.05$, FDR correction for six comparisons): (1) a low frequency cluster (gravity frequency, GF = 36.07±5.38 Hz, n = 71, *Table 1*), which is denoted as slow gamma (S-gamma). (2) a medium-frequency cluster (GF = 99.12±17.15 Hz, n = 71, *Table 1*) denoted as medium gamma (M-gamma), (3) a high-frequency cluster (GF = 127.72±11.28, n = 71, *Table 1*) with a preferred phase early in the theta cycle (-2.57±0.83 radians, n = 71, *Table 1*) denoted as early fast gamma (EF-gamma) and (4) a high-frequency cluster (GF = 131.83±8.80, n = 71, *Table 1*) with a preferred phase late in the theta cycle (2.12±0.77 radians, n = 71, *Table 1*) denoted as late fast gamma (LF-gamma). The phase-difference between EF- and LF-gammas was around one quarter of a theta cycle when one considers that phase is cyclic.

The clustering of TG states described above was done for recordings during awake behavior. We then repeated the same analysis for data recorded from REM periods independently. We also extracted four TG states during REM periods from all rats (*Figure 1G*). The gravity frequency and theta phase were comparable with the states we found in awake periods (*Table 1*). Thus, we found four similar TG states during both theta dominated awake behavior and REM sleep.

We next wondered how well each theta cycle fit into these different TG states. To address this, we determined how similar each single theta cycle was to each TG state cluster and if theta cycles could be similar to more than one cluster. We computed the correlation between the FPP of each theta cycle and the mean FPP of theta cycles of the same state (intra-cluster correlation) or the other states (inter-cluster correlation). Almost all intra-cluster correlations were higher than the maximum

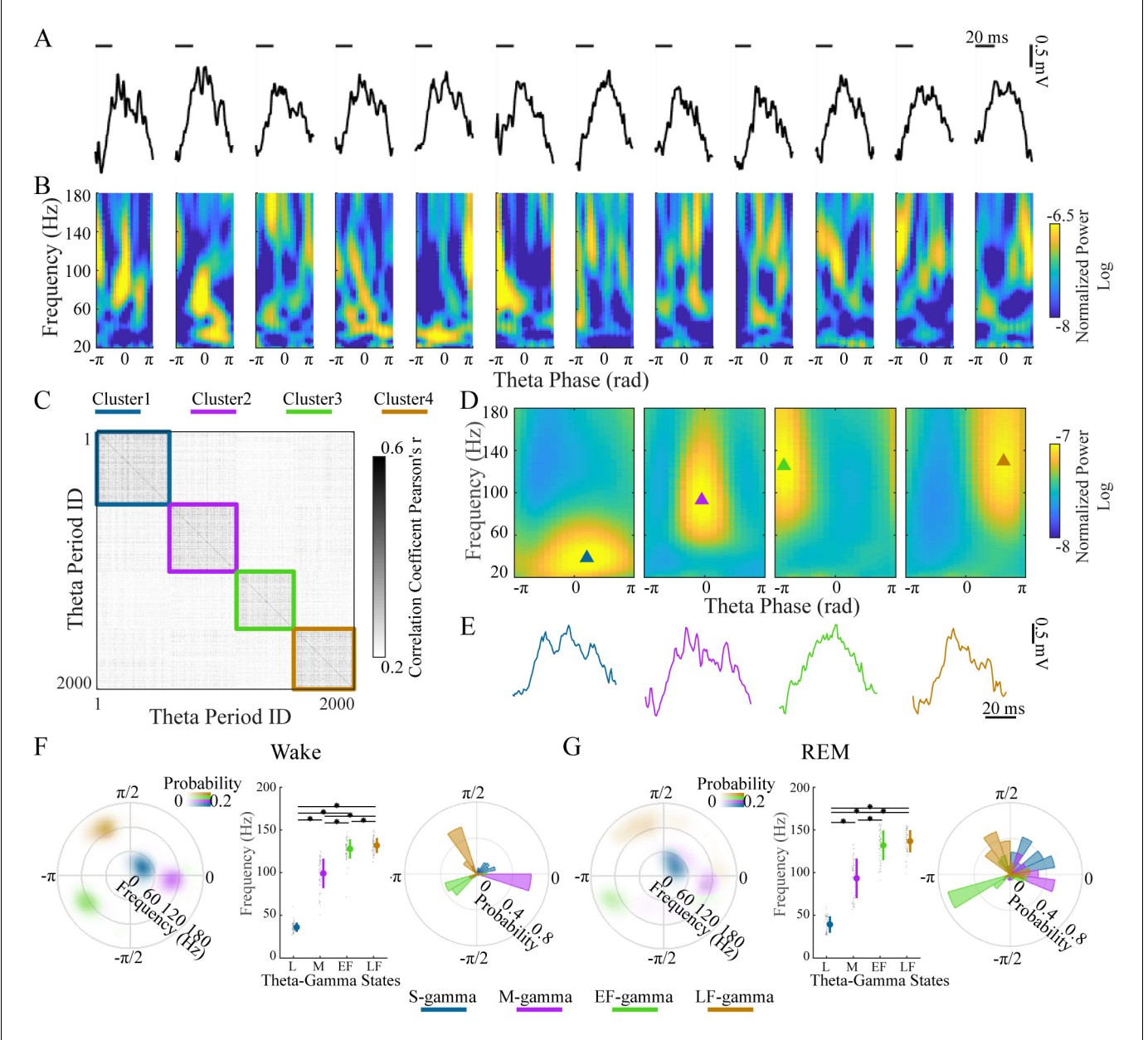

**Figure 1.** Clustering individual theta cycles based on cross-frequency coupling in hippocampal CA1. (**A**) A raw LFP recording trace with twelve successive theta cycles from CA1 in an awake rat. (**B**) FPP for each theta cycle in A. (**C**) Correlation matrix of 2000 FPPs, organized based on k-means clustering. (**D**) Average FPPs across different theta cycles within the four clusters (n = 13720, 11404, 10745 and 14104 theta cycles from top to bottom) from one rat (Cicero, S09102014) during awake periods. Triangles indicate the center of gravity (see Materials and methods). (**E**) Individual example LFP traces for the four TG states, respectively. (**F**) Density plot of the frequency and theta phase for the center of gravity of the four types of gamma fields (fields were defined as 95% of peak value and above) detected from all rats during awake periods, left; frequency for the center of gravity (mean ± sd, n = 71 channels from 9 rats), middle, * denotes significant difference (paired t-test, q < 0.05, FDR correction for multiple comparisons); the distribution of theta phase for the center of gravity for the four TG states, right. (**G**) As in F from all rats during REM periods (mean ± sd, n = 64 channels from 8 rats). FPP: frequency and theta phase power. sd: standard deviation. TG: theta-gamma.

DOI: https://doi.org/10.7554/eLife.44320.002

The following figure supplements are available for figure 1:

**Figure supplement 1.** LFP recordings at different recording depths.

DOI: https://doi.org/10.7554/eLife.44320.003

**Figure supplement 2.** Clustering of wavelet power in frequency and theta phase domains.

DOI: https://doi.org/10.7554/eLife.44320.004

*Figure 1 continued on next page*

*Figure 1 continued*

**Figure supplement 3.** Averaged current source densities across channels for each TG cluster.
DOI: https://doi.org/10.7554/eLife.44320.005
**Figure supplement 4.** Intra-cluster sample correlation versus inter-cluster sample correlation.
DOI: https://doi.org/10.7554/eLife.44320.006
**Figure supplement 5.** Cross-validation for individual theta cycle assignments.
DOI: https://doi.org/10.7554/eLife.44320.007
**Figure supplement 6.** Clustering of tetrode data.
DOI: https://doi.org/10.7554/eLife.44320.008

inter-cluster correlations (the highest of the three inter-cluster correlations for each theta cycle; see Materials and methods) and on more than 50% of theta cycles intra-cluster correlations were 0.15 higher than the maximum inter-cluster correlations (*Figure 1—figure supplement 4*). In other words, the FPP of those theta cycles were much more similar to its TG state than the other three states. This suggests that most, but not all, theta cycles could be clearly classified into one specific TG state. Around 20% of theta cycles had a difference between intra- and maximum inter-cluster correlation that was close to zero, specifically less than 0.05 (*Figure 1—figure supplement 4*), showing that those samples share some similarities with at least two TG states. This analysis was performed with five-fold cross validation (see Materials and methods). Overall, our results suggest that the four TG states are distinct from each other, while some individual theta cycles have features of multiple TG states.

We then performed a cross-validation across channels within the same animals and across animals to understand how clustering differed across recordings. Within the same session or animal, we selected signals from one given recording channel as a training channel and the others (different channels within the same animal or different animals) as testing channels. For the training channel, we calculated average FPPs from within the same theta-gamma states as reference FPP (as in *Figure 1D*). We then calculated how similar each testing theta cycle is to this training data by computing the correlation between the testing FPP and the reference FPP. The test theta cycle was assigned to a theta-gamma state based on the reference FPP state with which it had the highest correlation. Consequently, this analysis produced a new theta-gamma coupling state assignment based on clustering from a different animal or channel. By comparing this new theta-gamma state assignment with the theta-gamma state determined based on clustering of the testing channel, we calculated the accuracy of predicting the theta-gamma state across channels or animals. The cross validation was in general above chance levels (25% because a given theta cycle could be in four states), but highly variable ranging from 0.30 to 0.96 (*Figure 1—figure supplement 5* top panel).

We also tested our approach on tetrode data (*Figure 1—figure supplement 6*). The frequency and theta phase features for the center of gravity of different clusters varied across tetrodes. We found that in many tetrode recordings, slow gamma occurred at the peak of theta (0 phase; *Figure 1—figure supplement 6*, purple rectangle) similar to what we found in data recorded from silicone probes and the four clusters appear similar to those found with probe data. However, in other

**Table 1.** Frequency and phase of gamma power fields for each TG state.
Frequency (mean ± sd) and phase (circular mean ± circular sd) for the centers of gravity of the mean FPPs for LFPs recorded from all animals in data sets Hc-11 and Hc-3 during awake (71 recordings in nine animals) and REM (64 recordings in eight animals) periods. sd: standard deviation.

| FPP cluster | Wake (n = 71 recordings) | | REM (n = 64 recordings) | |
| --- | --- | --- | --- | --- |
| | Frequency (Hz) | Phase (rad) | Frequency (Hz) | Phase (rad) |
| S-gamma | 36.07 ± 5.38 | 0.58 ± 0.51 | 39.28 ± 9.69 | 0.74 ± 0.53 |
| M-gamma | 99.12 ± 17.15 | −0.04 ± 0.88 | 93.36 ± 23.16 | −0.01 ± 0.96 |
| EF-gamma | 127.72 ± 11.28 | −2.57 ± 0.83 | 131.97 ± 17.31 | −2.67 ± 1.19 |
| LF-gamma | 131.83 ± 8.80 | 2.12 ± 0.77 | 136.80 ± 13.04 | 1.68 ± 0.88 |

DOI: https://doi.org/10.7554/eLife.44320.009

tetrode recordings, slow gamma occurred near the trough of theta and the four clusters only sporadically identified medium gamma clearly. These differences may be due to the exact location of the tetrode relative to the pyramidal layer. For probe recordings, data from all channels were used to select the channel with the highest ripple power for clustering. Since this step cannot be performed in the same manner for tetrode recordings, we recommend selecting a channel with slow gamma at the peak of theta for clustering analysis when using tetrode recordings.

## Rapid behavior-dependent theta-gamma state transitions

Because each TG-state is thought to reflect a different computational state and coupling among hippocampal subregions, we investigated when these states occur and how the hippocampal network transitions between them. Transitions between TG states have important implications for how the network switches between different computational regimes. For instance, the network may persist in a single state for many theta cycles or switch states from one theta cycle to the next. Treating each theta cycle as an individual event, we examined a series of theta cycles as Markov chains (*Figure 2A*; *Figure 2—figure supplement 1*). We examined state occurrences as well as transitions from the current state to the next state for both awake exploration and REM periods, respectively (*Figure 2—figure supplements 2* and *3*). First, we found rapid switches between different TG states with transition probabilities ranging from 0.15 to 0.36 and the probability of remaining in the same state ranging from 0.15 to 0.49 (*Figure 2B,C*; *Figure 2—figure supplements 2* and *3*). Thus, the network remained in the same state less than half of the time showing for the first time that CA1 undergoes rapid TG state shifting.

TG transition probabilities changed depending on the behavior state of the animals. We compared the occurrence of each state and state transition when animals ran in a novel linear or circular track to periods when animals were in their home cage before and after the track sessions during waking periods. We found the occurrence of S-gamma states as well as transitions from all states to S-gamma decreased when animals ran in a track compared to pre- and post-track awake periods (*Figure 2B*; *Figure 2—figure supplement 2AB*, the first column; $p < 0.001$, $F_{38, 266}$ (TG states transition parameters × session)=11.07, two-way ANOVA repeated measures; paired t-test, $q < 0.05$, FDR correction for 60 comparisons). In contrast, both the occurrence of and transitions to M-gamma and EF-gamma were significantly enhanced when animals ran in the track compared to pre- and post-track sessions (*Figure 2B*). Occurrences of M-gamma as well as S/M/EF-gamma→M-gamma transitions were enhanced (*Figure 2B*; *Figure 2—figure supplement 2AB*, the second column; paired t-test, $q < 0.05$, FDR correction for 60 comparisons). EF-gamma occurrence and EF-gamma→EF-gamma transition were also higher in the track (*Figure 2B*; *Figure 2—figure supplement 2AB*, the third column). S/M-gamma→EF-gamma transitions were significantly increased if using a weaker significance threshold (paired t-test, $q < 0.1$, FDR correction for 60 comparisons). Furthermore, we also calculated the state transitions during early, middle, and late trials by separating the track session trials into three sections with the same number of trials each. Over the course of the trials, the environment was initially very novel and became less novel over trials. However, there was no difference in the transition matrix and occurrence of TG-states over these early, middle, and late trials as the track became less novel ($p = 0.462$, $F_{2, 14}$ (course) = 0.82; $p = 0.112$, $F_{38, 266}$ (TG states transition parameters × course) = 1.32; two-way ANOVA repeated Measures; paired t-test, $q > 0.1$, FDR correction for 60 comparisons).

During REM periods, we compared the occurrence of each state and state transition in the home cage before and after navigation in a novel environment (note that there are no REM periods during the track session). We found no significant differences using a strong significance threshold (paired t-test, $q > 0.05$, FDR correction for 20 comparisons). Using a weaker significance threshold (paired t-test, $q < 0.1$, FDR correction for 20 comparisons), we found decreased S-gamma→S-gamma transitions (*Figure 2C*; *Figure 2—figure supplement 3B*; $p = 0.008$, $t_7 = 3.70$, paired t-test) and increased S-gamma→LF-gamma transitions (*Figure 2C*; *Figure 2—figure supplement 3B*; $p = 0.007$, $t_7 = -3.73$, paired t-test). Thus, few differences were observed except for decreased S-gamma to S-gamma transitions and enhanced S-gamma to LF-gamma transitions over the entire REM period.

REM sleep is reported to be important for memory formation and consolidation (*Boyce et al., 2016*; *Buzsáki, 1998*; *Diekelmann and Born, 2010*; *Peever and Fuller, 2017*; *Rasch and Born, 2015*; *Rasch and Born, 2013*); however, it is not clear whether there are dynamic changes over the course of REM sleep. Interestingly, at the beginning of REM both before and after track sessions,

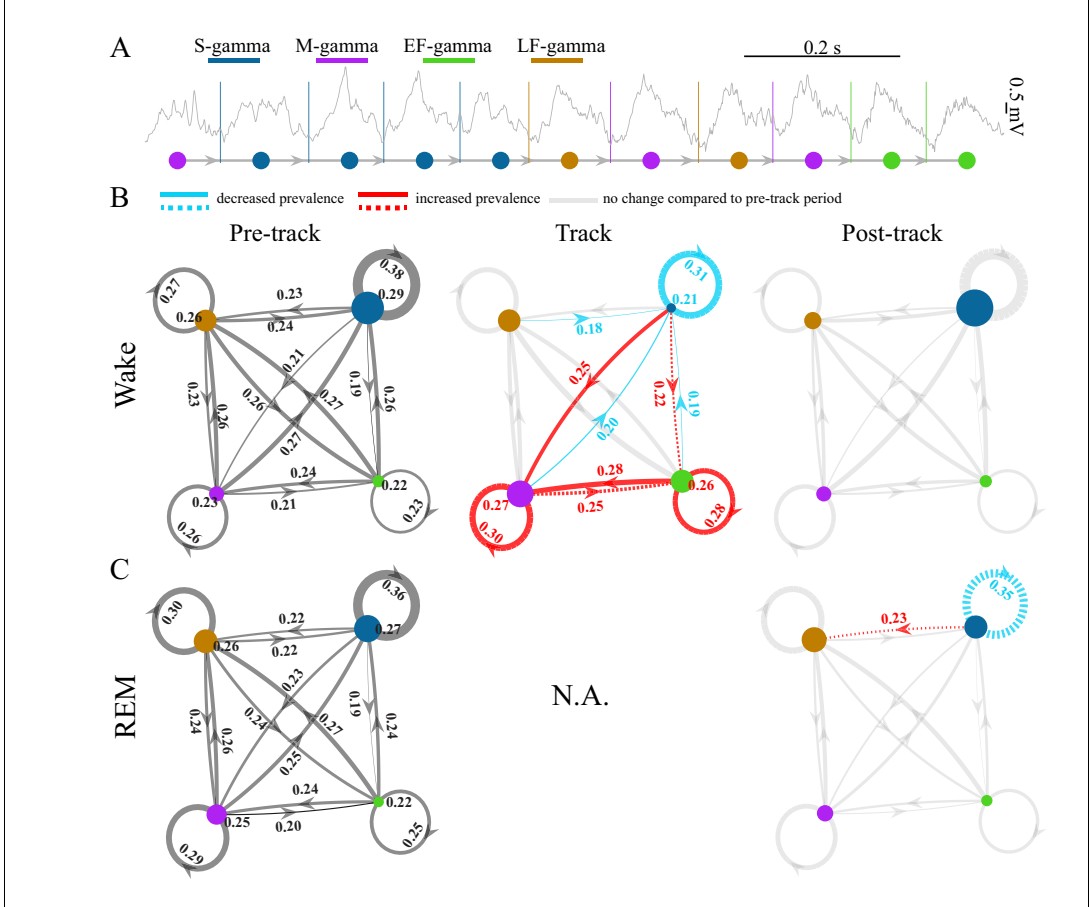

**Figure 2.** TG state transitions in awake and REM periods. (**A**) An example LFP trace is shown with each theta trough marked by vertical lines and an illustration of TG state transitions below. (**B**) Transition matrices and occurrence probabilities of the four states summarized during pre-maze (left), maze (middle) and post-maze (right) periods respectively for all rats during awake periods (n=8 sessions from 4 animals in Hc-11). Paired t-tests were performed across pre-, post-, and maze periods for 20 dynamic parameters including 16 (4 states × 4 states) transitions and 4 state occurrences. Only significant changes from the pre-maze period (paired t-test, q < 0.05, FDR correction for 60 comparisons) are highlighted and color coded in red (increased prevalence) and blue (decreased prevalence), less strict statistics are highlighted by a dashed line for q < 0.1. (**C**) The same as B for REM periods. Because there is no REM during maze exploration, pre and post maze comparisons were made. None of the 20 parameters reached significance (paired t-test, q > 0.05, FDR correction for 20 comparisons); S-gamma→S-gamma and S-gamma→LF-gamma are reduced and enhanced respectively by using less strict statistics (q < 0.1, FDR correction for 20 comparisons).

DOI: https://doi.org/10.7554/eLife.44320.010

The following figure supplements are available for figure 2:

**Figure supplement 1.** LFP examples during awake sessions.

DOI: https://doi.org/10.7554/eLife.44320.011

**Figure supplement 2.** Transition matrix and occurrence of the four TG states during awake periods.

DOI: https://doi.org/10.7554/eLife.44320.012

**Figure supplement 3.** Transition matrix and occurrence of the four TG states during REM.

DOI: https://doi.org/10.7554/eLife.44320.013

**Figure supplement 4.** Automatic behavior state detection.

DOI: https://doi.org/10.7554/eLife.44320.014

S-gamma occurred more frequently than the other states, persisting for approximately the first 20-30s of REM (*Figure 3*; p < 0.001, $F_{21, 609}$ (TG states × Time) = 4.69, two-way ANOVA repeated Measures). As time progressed, the occurrence of S-gamma decreased gradually, and stabilized. In contrast, the other gamma states were stable across time during REM (*Figure 3*). These novel results show that S-gamma is significantly higher than other theta-gamma coupling states during early REM.

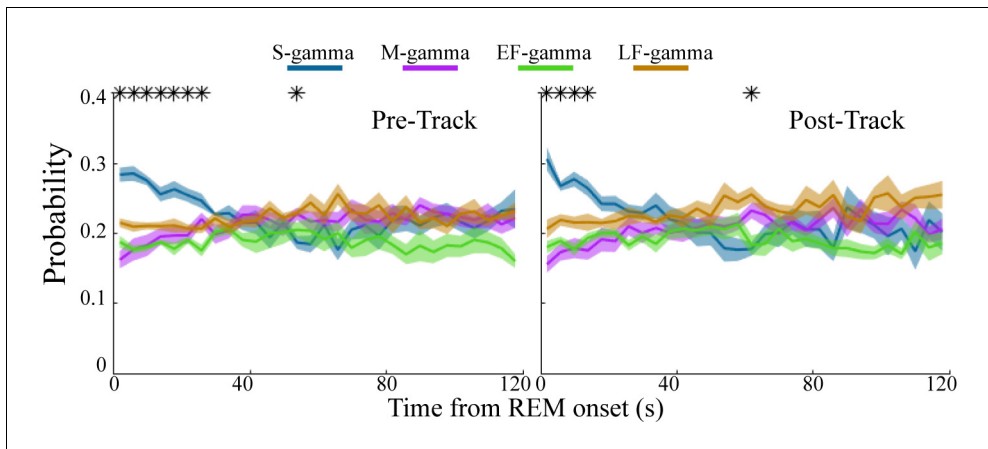

**Figure 3.** Low gamma dominates early REM. State occurrence rates were calculated in 4s bins starting from REM onset from 0 to 120s for pre- and post-maze periods respectively (mean ± sem, n=8 sessions from 4 animals in Hc-11). * represents significant differences found across the four TG states (one way repeated measures ANOVA, q < 0.05, FDR correction for 30 comparisons). sem, standard error of the mean.
DOI: https://doi.org/10.7554/eLife.44320.015

## Changes in CA1 coupling with CA3 and EC during different theta-gamma states

Different frequencies of gamma are theorized to couple-specific hippocampal subregions to support different functional states. Therefore, if the TG state transitions we found indicate switching between different functional states, then we would expect these states to have different coupling with other hippocampal subregions. Accordingly, we next calculated coupling of LFPs between CA1 and CA3 or between CA1 and EC and how they differed between these four TG states during awake periods. To achieve this, we calculated the pair-wise phase consistency (PPC) between LFPs recorded from CA1 and EC or CA1 and CA3 on each theta cycle using wavelets, as done previously (*Rohenkohl et al., 2018*; *Vinck et al., 2012*), for dual recordings in CA1 and CA3 (five sessions from two animals) or CA1 and EC (five sessions from three animals) from the Hc-3 data set (see Materials and methods). During exploration, we found a dominant peak in CA1-CA3 PPC in the slow gamma band (20-50 Hz) only during S-gamma states (*Figure 4A* top left; B left, blue arrow; p < 0.001, $F_{213, 17040}$ (TG states × Frequency) = 44.80, two-way ANOVA repeated measures; paired t-test, q < 0.05, FDR correction for 486 (81 Frequencies × 6 states pairs) comparisons), which was not present in CA1-EC PPC (*Figure 4A* top right; B middle and right). In contrast, CA1-EC PPC showed a dominant peak in the medium gamma band (60-120 Hz) in M-gamma states in some recordings (*Figure 4A* , top right). Furthermore, combined data across all recordings revealed higher CA1-EC PPC above 70Hz in M-, EF-, and LF-gamma states than in the S-gamma state (*Figure 4B* middle, p < 0.001, $F_{48, 3840}$ (TG states × Frequency) = 29.17, two-way ANOVA repeated measures; and *Figure 4B* right, p < 0.001, $F_{78, 6240}$ (TG states × Frequency) = 40.13, two-way ANOVA repeated measures; paired t-test, q < 0.05, FDR correction for 486 comparisons). Particularly, CA1-EC PPC was highest for the 60-80 Hz band and the 100-140 Hz band for M-gamma and EF-gamma states, respectively (*Figure 4B* , middle and right, purple and green arrows).

As during awake behavior, CA1-CA3 PPC during REM was the highest in S-gamma states in the slow gamma band (*Figure 4C* left, p < 0.001, $F_{21, 1680}$ (GT states × Frequency) = 10.48, two-way ANOVA repeated measures; paired t-test, q < 0.05, FDR correction for 486 comparisons), however it is important to note that this data is drawn from one session in one animal. CA1-EC PPC (four sessions from three animals) showed peaks in 40-70 Hz within the M-gamma state (*Figure 4C* top right), which was higher than the other three states although it was not significant (*Figure 4D* right, purple arrow).

To summarize, S-gamma states had the strongest CA3-CA1 PPC in the slow gamma band in both REM and awake periods. In contrast, EC-CA1 PPC was strongest in M-gamma states and EF-gamma states in the medium and high gamma bands, respectively. EC-CA1 PPC also showed the lowest

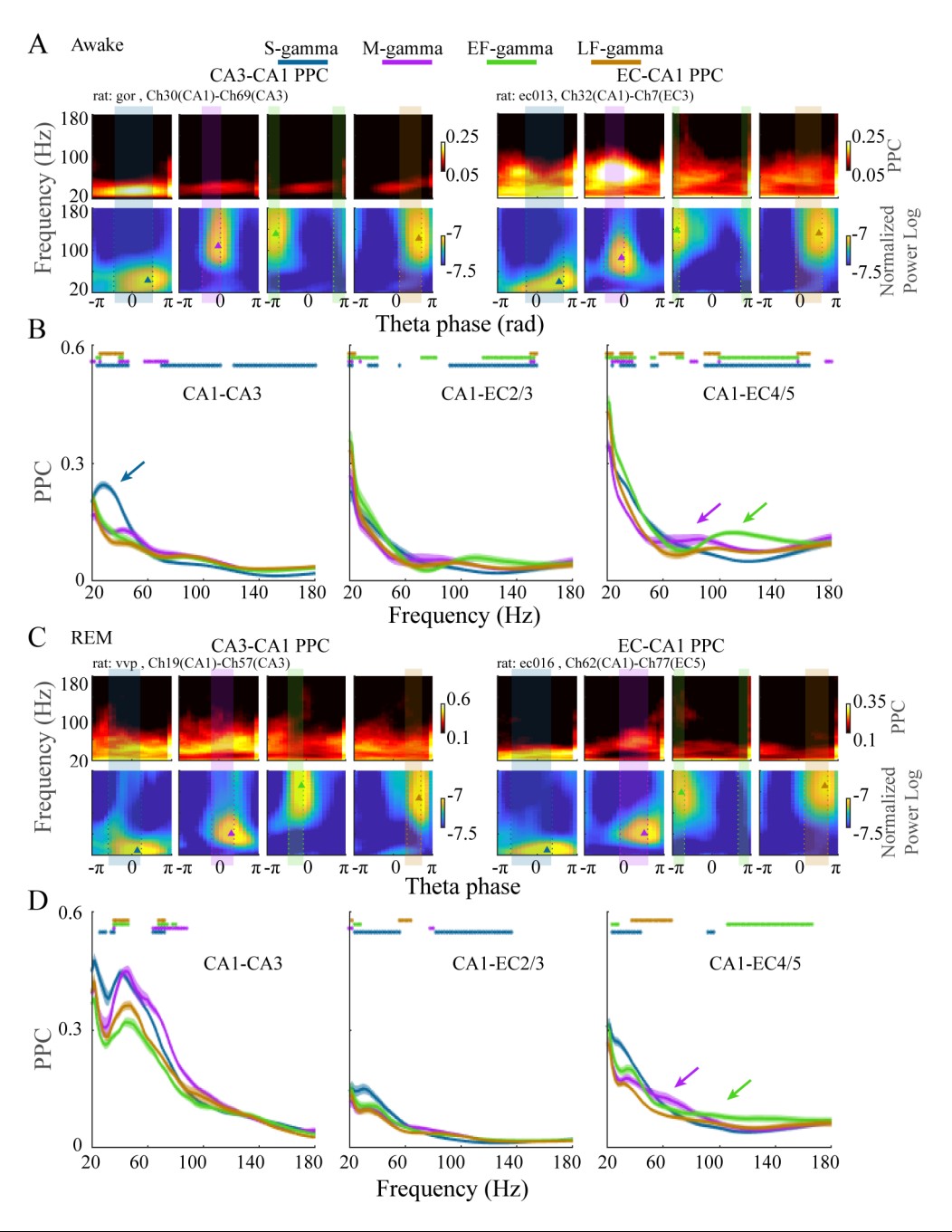

**Figure 4.** EC-CA1 and CA3-CA1 phase synchrony in the four TG states during awake and REM periods. (**A**) Examples of CA3-CA1 pairwise phase consistency (PPC, top left) and EC-CA1 PPC (top right) as a function of frequency and theta phase for each TG state during awake periods. The corresponding CA1 spectrum is shown below and the highlighted region (translucent rectangle) was used to calculate the average PPC between LFPs (see Materials and methods). (**B**) Average PPC (mean ± sem, n=72 CA1-CA3 channel pairs from five sessions in two animals, n=17 CA1-EC2/3 channel pairs from five sessions in three animals, n=27 CA1-EC4/5 channel pairs from five sessions in three animals) within the highlighted theta phase interval in A across animals during awake periods. PPC was compared across TG states across frequencies (81 frequency samples from 20 to 180 Hz), TG states that were significant from all other states were highlighted by the corresponding color bar above (paired t-test, q < 0.05, FDR correction of 486 = 6 states pairs × 81 Frequency sample comparisons). (**C**) As in (**A**) but during REM periods. (**D**) As in (**B**) but during REM periods (n=8 CA1-CA3 channel pairs from one session in one

*Figure 4 continued on next page*

*Figure 4 continued*

animal, n=34 CA1-EC2/3 channel pairs from four sessions in three animals, n=22 CA1-EC4/5 channel pairs from four sessions in three animals). FDR: false discover rate. sem: standard error of the mean.
DOI: https://doi.org/10.7554/eLife.44320.016

values in medium and high gamma band during S-gamma states during awake and REM periods. Together, these results show that these four TG states in CA1 differ in their phase synchrony with CA3 and EC. The stronger CA3-CA1 synchrony during S-gamma and EC-CA1 synchrony during M-gamma are comparable with previous findings (*Colgin et al., 2009*), that originally identified slow and medium gammas (which were called slow and fast gammas in that publication). However, stronger EC-CA1 synchrony during EF-gamma and the difference between EF- and LF-gamma synchrony has not previously been reported.

## Spiking during S-gamma has lower spatial information and phase precession

Different types of gamma are hypothesized to reflect different hippocampal functions and previous work has shown different spatial coding properties during slow and medium gamma (*Amemiya and Redish, 2018*; *Bieri et al., 2014*; *Colgin et al., 2009*; *Fernández-Ruiz et al., 2017*; *Zheng et al., 2016*). Thus, we wondered if hippocampal firing patterns or spatial coding differed across the four TG states we identified. First, we characterized firing rates in each gamma state to determine if spiking was equally distributed across different states, and then we characterized spike-field phase synchrony to determine if phase modulation was similar across different states. Neurons fired at lower rates in the S-gamma state for both interneurons and putative pyramidal cells (*Figure 5—figure supplement 1*). Across all theta cycles, interneurons showed significantly higher spike-field phase synchrony than pyramidal cells in the theta band as well as slow and medium gamma bands, represented by their higher spike-LFP PPC values (*Figure 5A*); while pyramidal cells showed higher PPC values in the high gamma band (*Figure 5A*). When we calculated PPCs for each TG state, we found pyramidal cells showed almost no difference in spike-LFP PPCs across states (*Figure 5—figure supplement 4*). In contrast, interneurons had significantly higher PPCs in S-gamma states in the slow gamma band (*Figure 5B*) and we found significant differences in interneurons' PPCs in the medium gamma band across TG states (*Figure 5B*).

We next tested whether place cell activity differed across the four TG states. We found that spatial information was similar across M-, EF-, and LF- gamma periods but lower during S-gamma. We examined the firing properties of place cells across states. In general, place cells fired at lower rates ($1.19 \pm 0.91$ Hz) in S-gamma states (*Figure 5—figure supplement 1*). Peak firing rate ($p<0.001$, $F_{3, 423}$ (TG states)=65.75, one-way ANOVA repeated measures) and spatial information ($p<0.001$, $F_{3, 423}$ (TG states)=30.99, one-way ANOVA repeated Measures) were also lower in the S-gamma state (*Figure 5CD*; paired t-test, $q < 0.05$, FDR correction for six comparisons). We also calculated the above parameters when animals traveled at different speeds. We found more S-gamma when animals did not move (*Figure 5—figure supplement 2A*). Furthermore, animal speed did not seem to account for differences in cells' firing properties across TG states (*Figure 5—figure supplement 2*). Additionally, we found low spatial information (<1 bits/theta cycle with 67% of having <0.1 bits/theta) in the occurrence of TG state events, for periods when animals ran faster than 5 cm/s, the same criteria used for calculating the spatial information of place cells (*Figure 5—figure supplement 3*). These results show that the spatial preference of TG states is much lower than that of place cells (>1.2 bits/spike for more than 90% of the cells). Furthermore, these results show that S-gamma has lower spatial information and is more likely to occur when animals are moving slowly.

We then examined phase precession in each TG state because previous reports suggest phase precession is under dual entorhinal and CA3 control (*Amemiya and Redish, 2018*; *Fernández-Ruiz et al., 2017*). First, we calculated the phase-precession for spikes that fired in each TG state by calculating phase-position regression and correlations across spikes that occurred during that TG state (*Figure 5E*, top). Generally, because separating spikes into their respective TG states reduced the number of spikes, the phase-position regression slope ($p=0.002$, $F_{4, 564}$ (gamma states)=4.33, one way ANOVA repeated measures; paired t-test, $q < 0.05$, FDR correction for 10 comparisons;)

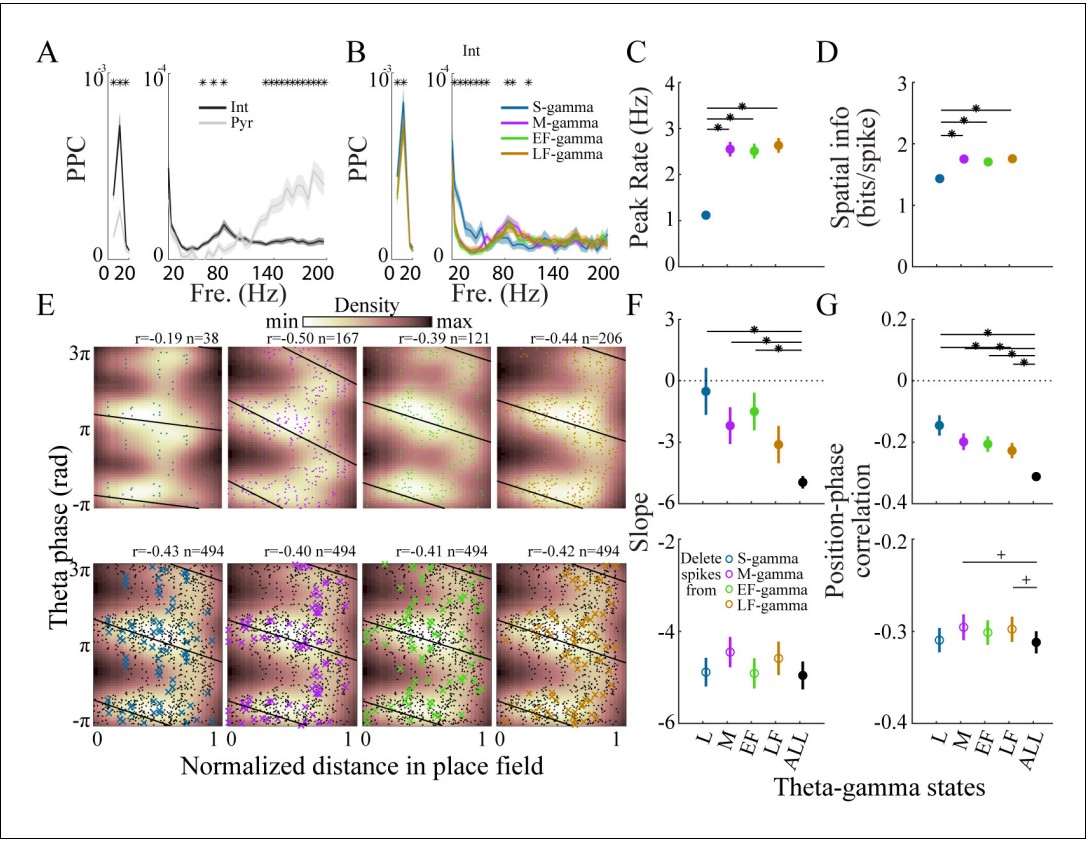

**Figure 5.** Interneurons and place cell activity in the four TG states. (**A**) Spike-field pairwise phase consistency (PPC) for interneurons (n=100) and pyramidal cells (mean ± sem, n=266), * represents significant cell-type differences (one-way repeated measures ANOVA, q < 0.05, FDR correction from 33 comparisons). (**B**) Spike-field PPC of interneurons in the four TG states. * indicates significant differences across TG states (one way repeated measures ANOVA, q < 0.05, FDR correction from 33 comparisons). (**C**) Peak firing rate of place cells (n=142) for different TG states. * indicates significant differences (paired t test, q < 0.05, FDR correction for six comparisons). (**D**) Spatial information of place cells (n=142) for different TG states. * indicates significant differences (paired t test, q < 0.05, FDR correction for 6 comparisons). (**E**) Phase precession of an example unit (Unit 1008, rat Achilles S11012013) for four different TG states, thick black line shows phase-position regression (top). Phase precession of the same unit after randomly deleting 38 spikes (represented by ×), the minimum number of spikes of the four states, from each TG state (bottom). The place field entry and exit were normalized to 0 and 1, respectively, on the x axis; the phase-position correlation coefficient, r, is shown above each figure. (**F**) Slope of phase-position regression (n=142 fields) for each TG state (top panel) and after deleting spikes for each TG state (for each unit, results are averaged from 100 random deletions). Black dots indicate measures from including all spikes. * indicates significant differences (paired t test, q < 0.05, FDR correction for 10 comparisons); + indicates comparison reached significance threshold of q < 0.1. (**G**) As in (**F**) for phase-position correlation of the 142 place fields. sem, standard error of the mean.

DOI: https://doi.org/10.7554/eLife.44320.017

The following figure supplements are available for figure 5:

**Figure supplement 1.** Firing rates of single units during the four TG states in awake periods.
DOI: https://doi.org/10.7554/eLife.44320.018
**Figure supplement 2.** Neural firing properties across different animal speeds.
DOI: https://doi.org/10.7554/eLife.44320.019
**Figure supplement 3.** TG event occurrence as a function of animal spatial position.
DOI: https://doi.org/10.7554/eLife.44320.020
**Figure supplement 4.** Pairwise phase consistency of pyramidal cells in the four gamma states.
DOI: https://doi.org/10.7554/eLife.44320.021

and phase-position correlation (p<0.001, $F_{4, 564}$ (gamma states)=7.73, one way ANOVA repeated measures; paired t-test, q < 0.05, FDR correction for 10 comparisons) were lower in each state than when all spikes in all states were included (*Figure 5FG*, top). Comparing the TG states to each other, phase precession in S-gamma states showed a trend of being weaker than the other three, although it was not significantly different (p>0.05, paired t-test) except for the phase-position correlation with LF-gamma (*Figure 5G*, top).

To control for differences in the number of spikes in each state, we deleted spikes from each gamma state until all states had the same number of spikes and re-calculated phase precession (*Figure 5E*, bottom). The number of spikes deleted was determined by the lowest spike population in the four states, which was usually S-gamma. The same number of spikes was deleted randomly 100 times and phase precession was calculated using the remaining spikes, then the results were averaged across the 100 deletion cases. For the slope of phase precession, after controlling for spike counts, no difference (p=0.14, $F_{4, 564}$ (gamma states)=1.75, one way ANOVA repeated measures) was found in S-gamma (p=0.71, $t_{141} = 0.38$ paired t-test), EF-gamma (p=0.73, $t_{141} = 0.35$ paired t-test) and LF-gamma (p=0.22, $t_{141} = 1.23$ paired t-test) compared to raw values in which no spikes were deleted. Deleting spikes from M-gamma states did slightly increase the slope value (p=0.02, $t_{141} = 2.42$ paired t-test) though the change was not significant (*Figure 5F*, bottom; q > 0.05, FDR correction for 10 comparisons). Controlling for spike counts significantly affected phase-position correlations (p=0.04, $F_{4, 564}$ (gamma states)=2.48, one-way ANOVA repeated measures). Phase-position correlations were significantly higher in M-gamma (p=0.02, $t_{141} = 2.42$ paired t-test; q < 0.1, FDR correction for 10 comparisons) and LF-gamma (p=0.01, $t_{141} = 2.54$ paired t-test; q < 0.1, FDR correction for 10 comparisons, *Figure 5G*, bottom). These results show that spikes from M and LF-gamma made a significant contribution to theta phase precession. Thus, we show for the first time that place cell firing in M-, EF-, and LF-gamma states contributed to spatial tuning significantly more than in S-gamma; while M- and LF-gamma contributed to phase precession significantly more than other states.

## Discussion

Here, we describe a novel method to classify individual theta cycles into distinct theta-gamma coupling states by combining signal processing and machine learning. We investigated theta-gamma coupling in individual theta cycles based on both frequency and theta phase features in an assumption-free manner. Leveraging the ability to classify every theta-cycle, we calculated inter-theta states transitions for the first time. We identified four distinct TG states and found that they dynamically changed from one cycle to the next. The rapid switching between TG states coupled with distinct CA3-CA1 and EC-CA1 coherence in different states supports the theory that theta oscillations facilitate rapid changes in information flow through the hippocampal circuit. We also show distinct interneuron spike-field coherence and pyramidal cell spatial coding in different TG states. Together, these rapid changes in theta-gamma coupling states from one theta-cycle to the next coupled with the distinct coherence and neural coding of different TG states shows that the HPC rapidly shifts between distinct functional states.

### Single-cycle classification of theta-gamma coupling based on frequency and phase

To cluster single theta cycles into different states, we used k-means clustering with k (the number of clusters) determined by community clustering. K-means clustering, which is a partially supervised learning method, gave robust clustering results across str. pyr. The identified TG state for each theta cycle was highly correlated across all recording depths, revealing that this method is robust regardless of the exact electrode position within the CA1 stratum pyramidale (MATLAB code to perform this analysis is provided, see Materials and methods). Clustering was performed on datasets recorded from the CA1 pyramidal layer (Hc-3 and Hc-11). These recordings spanned 160 or 200 μm targeted to the pyramidal layer and may include some areas above and below the pyramidal layer but did not cover all input layers of CA1. We found highly comparable results from two different datasets suggesting that this approach is consistent and robust in processing LFPs recorded from the pyramidal layer of hippocampal CA1 from rats. However, clustering results from other layers, regions or species has yet to be determined. Because we found variability in cross validation

analyses across animals and channels, we recommend others apply our method to identify TG states in their own data directly instead of using the TG states we identified as a standard. For tetrode recordings, we found some variation in clustering across channels with similar clusters observed on channels that had slow gamma at the peak of theta. Thus, we recommended selecting such a channel with slow gamma at the peak of theta for clustering analysis when using tetrode recordings. Together, these results show that this is a robust method to track the dynamics of theta-gamma coupling. Importantly, this approach is readily applied to track dynamic changes of any type of cross-frequency coupling in any brain region or in other applications. Furthermore, this method can also identify the existence of sub-groups of other types of oscillatory events.

We examined how similar each theta cycle was to different TG states. Most theta cycles were highly correlated with just one TG state but some (~20%) individual theta cycles were correlated with multiple TG states. These results show that while most theta cycles fit into only one TG state, some theta cycles may have features of multiple TG states. As a result we conclude that these four TG states can mix or overlap within a single theta cycle.

Several recent papers have investigated hippocampal theta or theta-gamma coupling on a cycle-by-cycle basis (*Dvorak et al., 2018*; *Lopes-Dos-Santos et al., 2018*; *Zheng et al., 2016*). Dvorak et al. detected single gamma events directly in the pyramidal layer and revealed fluctuations in the ratio of slow and medium gamma occurrences (*Dvorak et al., 2018*). Similarly, we also show slow and medium gamma components are detected within the pyramidal layer and TG occurrences fluctuate over time. They further found that the ratio of slow to medium-gamma reaches a local maximum before animals have to avoid a part of the environment where they have previously been punished. Lopes-dos-Santos et al. classified single theta cycles based on their spectral components using frequency decompositions of LFP recordings and independent component analysis (ICA) and identified four theta-gamma components in the hippocampus (*Lopes-Dos-Santos et al., 2018*). Although they identified the same number (four) of theta-gamma components as we did, the frequency and phase of their four theta-gamma states differed. Specifically they identified a theta-beta component (with beta around 22 Hz) in addition to previously characterized slow, medium, and fast gammas. We identified two fast gamma states in addition to previously characterized slow and medium gammas. These differences may be due to different frequency bands included in analysis (10–200 Hz in their case and 20–180 Hz in our analysis) and different methods. We did not include 10–20 Hz in our analysis because that is generally considered out of the gamma range. Furthermore, their method would be unable to separate fast gamma into two states because their method does not take into account the theta phase at which the gamma oscillations occur. While their analysis is a significant step forward in examining theta-gamma coupling, it has several limitations. ICA makes strong theoretical assumptions of the existence of independent non-Gaussian sources in the system and the number of independent components. Furthermore, this approach results in negative powered components that are hard to interpret. More importantly, their ICA decomposition uses only frequency information within each single theta cycle while we use information in both frequency and phase domains. Using theta phase information is crucial for separating EF- and LF- gammas. Furthermore, phase information is thought to be a key component of frequency coupling. Thus, our approach could be more suitable for tracking theta-gamma coupling based on both frequency and theta-phase features. While others have investigated hippocampal theta or theta-gamma coupling on a cycle-by-cycle basis our method it the first to: (1) classify of theta-gamma coupling using both frequency and phase information and (2) characterize state transition between theta-gamma states.

It is important to note that our method depends on accurate estimation of theta phase. We found some 'medium' gamma theta cycles were misclassified into EF-gamma clusters on recordings from deep pyramidal layers, and this misclassification may be due to less accurate estimation of theta phase at that recording location. The separation of EF- and LF- gammas especially depends on accurate theta phase estimation because both EF- and LF- gammas occur near the theta trough and they have overlapping frequency content. However, EF- and LF- gammas differed in their coupling with EC. Thus, further investigation is needed to establish whether there are three or four distinct theta-gamma states in hippocampal CA1.

## Relationship to and extension of previously identified theta-gamma coupling

Our approach identified both previously identified TG states and new states. The S-gamma state we detected in both awake and REM periods, is dominant in the range of 30–50 Hz and nested in the descending phase of pyramidal theta, which is consistent with slow gamma reported previously (*Colgin et al., 2009*; *Schomburg et al., 2014*). Additionally, we found stronger CA3-CA1 coupling in 20–50 Hz in S-gamma states, as reported previously (*Colgin et al., 2009*; *Lasztóczi and Klausberger, 2016*; *Schomburg et al., 2014*). The M-gamma state we found falls into the frequency range of 60–120 Hz nested in the peak of pyramidal theta, which corresponds to medium gamma reported previously, first identified as fast gamma in *Colgin et al. (2009)* and later called medium or fast gamma (*Colgin, 2015b*; *Fernández-Ruiz et al., 2017*; *Lasztóczi and Klausberger, 2016*). We observed higher EC-CA1 coupling in 60–80 Hz or 40–70 Hz in M-gamma states than the other three states during awake periods and REM periods, respectively. These results agree with prior reports showing stronger CA3-CA1 synchrony during S-gamma and stronger EC-CA1 synchrony during M-gamma (*Colgin, 2015b*; *Fernández-Ruiz et al., 2017*; *Lasztóczi and Klausberger, 2016*). Finally, the EF-gamma and LF-gamma we identified fell into the >120 Hz range but occurred at different phases of theta with EF-gamma at the beginning of the theta ascending phase and LF-gamma at the late descending phase. Both EF- and LF-gamma are similar to fast gamma reported nesting in the trough of pyramidal theta but are quite distinct from each other in theta phase (*Amemiya and Redish, 2018*; *Fernández-Ruiz et al., 2017*; *Lasztóczi and Klausberger, 2016*). In addition, higher EC-CA1 coupling was observed in EF-gamma than in LF-gamma. Together, these results suggest that EF- and LF-gamma are distinct and that they are two sub-types of fast gamma not previously differentiated.

## Changes in internal states during behavior and sleep

Distinct patterns of LFP activity reflect different internal brain states in terms of the excitatory state of individual neurons, synchrony among neurons, and interactions between brain regions. These internal brain states are thought to reflect different computational states and therefore brain functions. However exactly how different brain functions map onto such states remains unclear, in part because our ability to detect such states is limited. Current signal processing methods to identify oscillatory states typically require many continuous oscillatory cycles and therefore have inadequate temporal resolution to detect rapid state changes. In the hippocampus, different gamma frequencies have been linked to different hippocampal functions including processing incoming sensory information, memory encoding, and memory retrieval. However, different studies have come to conflicting conclusions as to which type of gamma is related to which function (see *Colgin, 2015a* for review). Here, we describe a method that captures dynamic cycle-to-cycle changes in oscillatory coupling states. Using our approach, we calculated the dynamics of TG states at single-cycle resolution including state occurrence and transitions. This method will allow for more precise study of oscillatory states and therefore will help ascertain how these states relate to different brain computations and functions.

Generally, we found frequent cycle-to-cycle switching between different TG states showing that the hippocampus can rapidly shift between these different proposed functional states. These sub-second dynamic changes of TG states suggest that not only individual theta cycles but also cycle-by-cycle transitions should be further studied to understand hippocampal function.

Some hippocampal states are thought to prioritize the processing of external sensory information over memory retrieval. Stronger EC-CA1 than CA3-CA1 interactions are expected in such a state because EC is thought to provide CA1 with ongoing sensory information (*Bieri et al., 2014*; *Cabral et al., 2014*; *Newman et al., 2013*; *Takahashi et al., 2014*), while CA3 is believed to be essential for memory retrieval (*Bieri et al., 2014*; *Colgin, 2015a*; *Igarashi et al., 2014*; *Tort et al., 2009*). Furthermore, one would expect that processing sensory information would be especially important when animals explore a novel environment. Consistent with these studies, we found M-gamma but not S-gamma fit with these expected roles in sensory processing with dominant EC-CA1 coupling in the medium gamma band. We found significantly decreased S-gamma and increased M- and EF-gamma, supporting the notion that M- and EF-gamma but not S-gamma reflect sensory processing that is expected during exploration of a novel environment. During waking, CA1-

EC4/5 coupling was significantly stronger in the high gamma band during EF-gamma and was significantly weaker during S-gamma than other states, with M- and LF-gamma in between. These results could arise from stronger CA1-EC4/5 coupling during EF-gamma or they could indicate that EF- and LF-gamma undergo different transformations within CA1. Furthermore, we found lower spatial information during S-gamma than during other states; and S-gamma spikes made no significant contribution to theta phase precession, while M- and LF-gamma did. Together, these results suggest a role for M- and EF- and LF-gamma in this sensory information processing. However, we also found significantly higher CA3-CA1 coupling in the slow gamma band during both S- and M-gamma than during other states with higher coupling in S- than in M-gamma. These results suggest that M-gamma may also play a role in CA3-CA1 interactions, which has not been reported previously (*Cabral et al., 2014*; *Colgin, 2015a*; *Newman et al., 2013*; *Takahashi et al., 2014*; *Tort et al., 2009*).

Prior work revealed that as animals run faster, the frequency of gamma oscillations also tends to be faster (*Ahmed and Mehta, 2012*; *Colgin, 2015a*; *Kemere et al., 2013*). Consistent with these papers, we observed S-gamma was more likely to occur when animals moved slowly. Kemere et al. further showed that the relationship between gamma power and animal speed was stronger in a novel environment than in a familiar environment (*Kemere et al., 2013*). While, we did not explicitly examine the relationship between TG state occurrences and animal speed in novel versus familiar environments, we did detect some non-significant differences between novel and familiar environments. We observed trends of lower occurrences of S-gamma (p=0.18, $t_7 = -1.47$, early vs middle) and higher occurrences of M-gamma (p=0.08, $t_7 = 2.05$, early vs middle) in early trials, when the environment was more novel, although these differences were not significant.

While theta-gamma coupling has often been studied during waking and spatial navigation, less is known about REM (*Fernández-Ruiz et al., 2017*; *Montgomery et al., 2008*; *Schomburg et al., 2014*). This is especially important because REM sleep, during which theta predominates, is important for memory consolidation (*Boyce et al., 2016*; *Diekelmann and Born, 2010*; *Grosmark et al., 2012*; *Louie and Wilson, 2001*). Here, for the first time, we show that S-gamma is significantly higher than other TG states during early REM. This could point to a role for S-gamma in memory consolidation or homeostasis (*Borbély, 1982*; *Borbély et al., 2016*; *Tononi and Cirelli, 2014*; *Tononi and Cirelli, 2003*; *Watson et al., 2016*), as both are hypothesized functions of REM sleep. While S-gamma dominated during early REM, we also observed slightly decreased S-gamma to S-gamma transitions and enhanced S-gamma to LF-gamma transitions over the entire REM period after navigation in a novel environment. These interactions of S-gamma and LF-gamma could be related to synapse re-scaling according to the synaptic homeostasis hypothesis (*Borbély, 1982*; *Borbély et al., 2016*; *Tononi and Cirelli, 2014*; *Tononi and Cirelli, 2003*).

## Different theta-gamma states have distinct neural spiking and theta phase precession

Inhibitory interneurons participate in the generation of gamma oscillations widely in the brain (*Buzsáki and Wang, 2012*). We found distinct spike field phase synchrony patterns in interneurons during different TG states, indicating that these cells are differentially modulated in different gammas. Separating each TG state, interneurons were more strongly modulated at 20–60 Hz during S-gamma states and at 60–120 Hz during M-, EF-, and LF-gamma states, while pyramidal cell modulation did not vary significantly from one TG state to another. These differential effects on interneuron and pyramidal cell firing may be because interneurons are more strongly driven by gammas from CA3 and EC, known local gamma generators (*Fernández-Ruiz et al., 2017*; *Lasztóczi and Klausberger, 2016*; *Lasztóczi and Klausberger, 2014*; *Somogyi et al., 2014*).

We also examined how different TG states contributed to theta phase precession. While phase precession has been well characterized over many theta cycles (*Dragoi and Buzsáki, 2006*; *Huxter et al., 2008*; *O'Keefe and Burgess, 2005*; *O'Keefe and Recce, 1993*; *Schmidt et al., 2009*), few studies have focused on how phase precession varies across theta cycles. Recent studies suggest phase precession varies across single traversals of an environment or in different theta-gamma coupling states (*Amemiya and Redish, 2018*; *Zheng et al., 2016*). We found significantly weaker phase precession for spikes occurring in S-gamma than those in the other states and higher phase precession for spikes occurring in M- and LF-gamma. Together these results suggest that different TG states play distinct roles in hippocampal spatial coding.

For decades, many studies have examined hippocampal function in spatial navigation and learning and memory (*Buzsáki, 2005*; *Buzsáki et al., 2002*; *Buzsáki and Llinás, 2017*; *Eichenbaum, 2014*; *Grosmark et al., 2012*; *Harris et al., 2002*; *Hok et al., 2007*; *Huxter et al., 2008*; *Ito et al., 2015*; *Itskov et al., 2008*; *Kitamura et al., 2017*; *Kraus et al., 2013*; *Manns and Eichenbaum, 2009*; *McNaughton et al., 2006*; *Mizuseki et al., 2011*; *Montgomery and Buzsáki, 2007*; *Moreno et al., 2016*; *Moser et al., 2017*; *Moser et al., 2008*; *O'Keefe, 1976*; *O'Keefe and Dostrovsky, 1971*; *O'Keefe and Recce, 1993*; *Okuyama et al., 2016*; *Rolls, 2016*; *Rolls et al., 2005*; *Rolls and Wirth, 2018*; *Roy et al., 2017*; *Sirota and Buzsáki, 2005*; *Squire et al., 2015*; *Terrazas, 2005*; *Yamamoto and Tonegawa, 2017*). Precise spike timing relationships have been observed with different types of hippocampal oscillations over milliseconds to hundreds of milliseconds and this spiking timing can encode spatial sequences traversing seconds to minutes (*Carr et al., 2012*; *Davidson et al., 2009*; *Deng et al., 2016*; *Dragoi and Buzsáki, 2006*; *Dragoi and Tonegawa, 2011*; *Genzel et al., 2017*; *Grosmark et al., 2012*; *Gupta et al., 2010*; *Karlsson and Frank, 2009*; *Lee and Wilson, 2002*; *Montgomery et al., 2008*; *Pastalkova et al., 2008*; *Wilson and McNaughton, 1994*). However, prior research has not characterized single cycles of an oscillation nor their inter-event dynamics. Here, we proposed new methods to separate single theta cycles based on both frequency and phase information. Using this approach, we then investigated theta-gamma coupling dynamics. We found these different states were distinct in a variety of ways beyond frequency and phase content, including their interactions with CA3 and EC, prevalence during exploratory behavior and REM, spatial information, and theta phase precession. Our approach and these results will provide new perspectives to understand oscillatory states and hippocampal functions during different behaviors.

## Materials and methods

### Animals and data acquisition

Biological replicates in this work were defined as electrodes or experimental sessions or animals in different analyses. In total ten rats (Hc-11 and Hc-3 data sets) were included for analysis of probe recordings from two different public data sets produced in the Buzsaki lab and previously published (*Chen et al., 2016*; *Diba and Buzsáki, 2008*; *Grosmark and Buzsáki, 2016b*; *Mizuseki et al., 2009*); all data are available at https://crcns.org/data-sets/hc. Details are summarized in *Supplementary file 1*. These electrophysiological recordings used silicon-probes (NeuroNexus, Ann Arbor, MI) with 4 shanks and 8 sites per shank, 6 shanks and 10 sites per shank, or 8 shanks and 8 sites per shank. Probe recording sites were vertically staggered along the shank with 20 μm spacing between sites. Each site had an area of 160 μm2 and an impedance of 1–3 MΩ. Spike sorting was done by KlustaKwick (https://klusta.readthedocs.io/en/latest/) for automatic spike sorting, then by Klusters (http://klusters.sourceforge.net/) for manual adjustment. A third data set (Hc-19) recorded from one rat was used for testing our method on tetrode data.

### Hc-11 dataset

The Hc-11 data set is composed of 6- or 8-shank bilateral silicon-probe multi-cellular electrophysiological recordings performed on four male Long-Evans rats in the Buzsáki lab at NYU (*Grosmark and Buzsáki, 2016a*). These recordings were performed to assess the effect of novel spatial learning on hippocampal CA1 neural firing and LFP patterns in naïve animals. Each session consisted of a long (~4 hr) PRE rest/sleep epoch home cage recording performed in a familiar room, followed by a novel maze running epoch (~45 min) in which the animals were transferred to a novel room, and water-rewarded to run on a novel maze. All protocols were approved by the Institutional Animal Care and Use Committee of New York University.

### Hc-3 dataset

The Hc-3 data set is composed of 4- or 8-shank bilateral silicon-probe multi-cellular electrophysiological recordings performed on eleven male Long-Evans rats in the Buzsáki lab at Rutgers University (*Mizuseki et al., 2013*). Recordings were made in CA1, CA3, or entorhinal cortex (EC) of the right dorsal hippocampus. The individual silicon probes were attached to micromanipulators and moved independently. Only experiments that satisfied two criteria were included: (1) Dual recordings were

performed either in CA1 and CA3, or CA1 and EC and (2) animal behavior included either sleep sessions or linear maze sessions. All protocols were approved by the Institutional Animal Care and Use Committee of Rutgers University (protocol No. 90–042).

### Hc-19 dataset
The Hc-19 data set included one male Long-Evans rat with multiple tetrode recordings from right dorsal CA1 (*Ciliberti et al., 2018a*). Data were collected during pre-run, sleep, and post-run periods during a spatial navigation task (*Ciliberti et al., 2018b*). All protocols were approved by the KU Leuven (Leuven, Belgium) animal ethics committee and are in accordance with the European Council Directive, 2010/63/EU.

## Identification of non-REM, REM, and wake episodes
For the Hc-11 data set, rapid eye movement sleep (REM) and non-REM sleep (NREM) episodes were scored by the Buzsaki lab and described in *Grosmark et al. (2012)*, and *Grosmark and Buzsáki (2016b)*. In brief, REM and NREM periods were detected based on the LFP power ratio in theta (5–11 Hz) and delta (1–4 Hz) and electromyographic (EMG) signals. These periods were manually adjusted with visual inspection of whitened power spectra (using a low-order autoregressive model) and raw traces (*Mizuseki et al., 2011*; *Mizuseki et al., 2009*; *Sirota et al., 2008*). Falsely detected short segments were removed.

In the Hc-3 data set, EMG signals were not recorded but calculated from LFPs because EMG recordings have been reported to be highly correlated with intracranial derived LFPs (*Schomburg et al., 2014*). Behavior states (REM, NREM, wake) were detected with MATLAB based SleepScoreMaster (https://github.com/buzsakilab/buzcode/) (*Grosmark and Buzsáki, 2016b*). First, LFPs from each CA1 channel were converted into spectrograms and PCA was performed to separate different spectral components. The first principal component (PC1) of each spectrogram represented periods with power in low frequencies, specifically with strong differences between frequencies < 25 Hz and frequencies in the gamma range (40 Hz-100Hz). PC scores in PC1 were a bimodal distribution and a threshold at the distribution's trough was set to separate the NREM states (high PC1 scores) from all other behaviors. Similar methods using a cutoff at the minimum of the bimodal distributions were applied in both narrow band theta power ratio (5–10 Hz/2–20 Hz) and EMGs. High theta power and low EMGs represented REM period. Other states were then classified as arousal. Awake states were defined as arousal for at least 7 min. We also tested this method on the Hc-11 data set, and the results were comparable to traditional sleep scoring methods, described above (*Figure 2—figure supplement 4*).

## LFPs data selection
To isolate CA1 recorded signals from the pyramidal layer center, we computed the highest ripple power (root mean square of filtered LFPs in 150–250 Hz) among all recording channels within the same shank (*Figure 1—figure supplement 1*). This pyramidal layer channel was used for further analysis unless otherwise noted. Channels at top or bottom sites were excluded. In data recorded from CA3 and EC (from Hc-3) we used the best channel for slow wave and theta separation, which was identified by using SleepScoreMaster (https://github.com/buzsakilab/buzcode/) (*Grosmark and Buzsáki, 2016b*).

## Wavelet spectrum normalized by theta phase
To decompose each theta cycle into its time-frequency decomposition, LFPs were first down sampled to 625 Hz for faster subsequent calculations. Morelet wavelets were then applied to the LFP using the default setting of Morlet wavelet transform $\psi(\mathrm{x}) = e^{-x^2/2} \cos 5x$ in the MATLAB Wavelet toolbox (Mathworks, Natick, MA) to produce a time-frequency representation of LFP power. The wavelet power spectrum $WS(t,f)$ was smoothed $\pm 2$Hz in the frequency direction and $\pm$ 8ms in the time direction with boxcar smoothing around each local time-frequency point $(t,f)$. The wavelet power spectrum was z-scored across time for a given frequency. Instantaneous theta phase $\theta(t)$ was calculated using Hilbert transform on the LFP in the theta band (5-10Hz). We then extracted a time-frequency decomposition matrix for each individual theta

cycle $A_k = \{a_{t,f} = WS(t,f), \ t = T_k, \ T_k + 1, \ \cdots, T_{k+1}\}$, where $A_k$ is the $k$th theta period detected such that

$$\theta(T_k) = 0°, \ \theta(T_{k+1}) = 360°$$

$$\theta(T_k + p) < \theta(T_k + q), \ \textit{for any } p < q \textit{ and } T_k + q \leq T_{k+1}$$

As the duration of the theta cycle varies, we separated each theta cycle into 20 phase bins, thus $A_k$ was normalized into a 81 (*Frequency*) $\times$ 20 (*theta phase*) power (FPP) matrix, could be also denoted as $A_k(f,\theta)$. 20 phase bins were chosen for the theta phase analysis as this was sufficient to robustly extract phase features of the four gammas with acceptable computational demands. Note that, although Morlet wavelets are widely applied in LFP analysis, other wavelets such as Morse wavelets might produce better frequency and phase estimates.

## Clustering power in wide gamma band across theta cycles

To separate theta cycles into different theta-gamma coupling states, two clustering methods were applied in succession: community clustering and k-means clustering. Community clustering was employed to identify the exact number of clusters for subsequent k-means clustering in an unsupervised, data-driven fashion. K-means clustering was used for subsequent analyses because it was computationally more efficient than community clustering and it more reliably extracted medium gamma oscillations in the deep pyramidal layer of CA1. For clustering, the FPP for each theta cycle was considered as points in 1620 dimensional space (81 *frequencies* $\times$ 20 *phases*). A FPP matrix was also constructed to calculate current source density (CSD), defined as a CSD-FPP matrix by calculating wavelet power of CSD as a function of LFP theta phase. To facilitate the testing of pattern consistency between LFP and CSD in the phase-frequency of FPP, we averaged CSD-FPP matrices within clusters.

### Community clustering

Community clustering is an unsupervised method that identifies the number of clusters or communities in a scalable greedy fashion using principled heuristics. Based on graph theory, n nodes $A_k, \ k = 1, 2, \cdots, n$ of a graph are assigned into $c$ communities $\sigma_i \in \{1, 2, \ldots, c\}$; that is each node is assigned to a community $\sigma_i$, where $i = 1, 2, \ldots, n$. Q-modularity of a weighted graph is defined as the edge weights within the community minus the expected edge weights (*Leicht and Newman, 2008*); that is $Q = \frac{1}{m} \sum_{i,j} (B_{i,j} - p_{i,j}) \, \delta_{i,j}$, where $\delta_{i,j} = 1$ if $\sigma_i = \sigma_j$ and 0 otherwise; $p_{i,j} = k_i k_j / m$ represents the expected edge weight between vertex $i$ and $j$; $m$ is total the weight of all vertexes. $B$ is the adjacent matrix, where $B_{i,j}$ is the exact edge weight between node $i$ and node $j$. In this study, the adjacent matrix $B$ is defined as $B = C + 1$, where $C_{i,j}$ is the Pearson correlation between $A_i$ and $A_j$. Thus we avoid negative edge weights. Maximizing the Q-modularity produces the community structure with the densest intra-community connections and sparsest inter-communities connections. We applied the community detection algorithm to greedily maximize Q-modularity using the Louvain method (*Blondel et al., 2008*) with the corresponding MATLAB package (*Jeub et al., 2011*; http://netwiki.amath.unc.edu/GenLouvain).

Community clustering failed to detect medium gamma states in deeper parts of the pyramidal layer of CA1 (*Figure 1—figure supplement 2A* left panel). In such cases, community clustering may have misclassified 'medium' gamma theta cycles into the EF-gamma cluster based on the observation that two gamma fields are observed in this cluster (third column in *Figure 1—figure supplement 2A*, left panel), one in the high gamma range occurred at the ascending phase of theta and the other with lower frequency occurred at later theta phases. Therefore, we used community clustering to determine the appropriate number of clusters (four) in an unsupervised, data-driven fashion but used k-means clustering for more robust clustering across the pyramidal layer. K-means clustering was computationally more efficient than community clustering and more reliably extracted medium gamma oscillations in the deep pyramidal layer.

## k-means clustering

k-means clustering assigns the $n$ power samples $A_i$ in frequency-phase domain to exactly one of $k$ clusters defined by centroids, where $k$ is chosen before clustering (*Lloyd, 1982*). A distance matrix $D$ is used for clustering. Here, we use Pearson correlation distance $D = 1 - C$, where $D_{i,j}$ quantifies the distance between any pair of theta cycles $A_i$ and $A_j$. The Pearson correlation (c) is a standard similarity measure (cf. distance if 1-c) in image processing especially because it is invariant to power (*Kaur et al., 2012*). In using Pearson correlation as a distance measure for k-means, we are actually treating the FPP as a normalized image, and clustering based on notions of image similarity. It remains unknown how best to extract and quantify both frequency and phase features for theta-gamma coupling, especially considering that phase is circular. Note that in our analysis, phase wrapping does not affect the results. The correlation is computed from the same points or bins (in frequency and phase) between one image (FPP) and another. In other words, we are using Eulerian representation instead of Lagrangian representation (*Batchelor, 2000*). Therefore, the geometrical relationship between bins does not matter: if adjacent bins have similar values or those values are on opposite sides of the image does not affect the correlation. Here, we applied the k-means++ algorithm implemented in MATLAB Statistics and Machine Learning Toolbox (Mathworks, Natick, MA).

Gamma fields were defined as above a threshold of 95% of the peak of the average FPP within one cluster. The gravity frequency and theta phase for the center of gravity of each gamma fields was extracted. Thus, the feature of a specific cluster could be represented by the gravity frequency and gravity theta phase of its gamma field. Gravity frequency is defined as the weighted averaged of all frequencies (y coordinates) of points within the gamma field, while gravity phase is the weighted averaged (circularly) of all phases (x coordinates) of points within the gamma field. The weight of each point in the gamma field is defined by the power value at that point in the FPP. Sorting the gravity frequencies of the four clusters generated through k-means from one channel of LFP signal, the first and second lowest gravity frequencies corresponded to L- and M-gamma clusters, respectively. The higher two gravity frequencies were fast gammas, and were further separated as EF- and LF- gamma based on their gravity theta phase. The early/late notation was based on the phase of these high gammas relative to the phase of M-gamma. Our MATLAB code of the above protocol, including other MATLAB sub functions for clustering individual theta cycles with any LFP signals recorded from hippocampal CA1 region, can be found on our github (*Zhang, 2019*; copy archived at https://github.com/elifesciences-publications/IndividualThetaCluster).

## Intra-cluster correlation versus inter-cluster correlation

Intra-cluster correlation was the correlation value between a single theta cycle's FPP and the mean FPP of the state that theta cycle was assigned. The maximum inter-cluster correlation was the largest correlation value between a theta cycle's FPP and the mean FPP of every other state (not including the state that the theta cycle is assigned). The difference between intra-cluster correlation and max inter-cluster correlation (*Figure 1—figure supplement 4*) represented the difference of one given theta cycle between its assigning state and other states. Intra-cluster correlation, $\rho_{intra}$, and maximum inter-cluster correlation, $\rho_{max-inter}$, were defined as follows:

$$\rho_{intra}(i) \triangleq corr\left(FPP_i,\ FPP^k\right), j_i = k$$

$$\rho_{max-inter}(i) \triangleq max\left(\left\{corr\left(FPP_i,\ FPP^k\right)\right\}\right), j_i \neq k.$$

where $FPP_i$ denotes any given theta cycle in the training set (sample size N), clustered in state $j_i$, where $i = 1, 2, \ldots, N$ and $j_i \in \{1, 2, 3, 4\}$. The mean FPP for the four states was denoted as $FPP^k$, $k = 1, 2, 3, 4$. $corr$ indicates Pearson correlation. We performed a five-fold cross validation, where $FPP^k$ was calculated from training data sets and the sample $FPP_i$ was from test data sets with 5 repetitions. Similar results were found with and without cross-validation (data not shown).

## Cross-validation for individual theta cycle assignment

Within the same session or animal, we selected signals from one given recording channel as a training channel and the signals from different channels in the same animal or channels from different animals as testing channels to perform cross-validation within (intra-animal) or across (inter-animal)

animals, respectively. For the training channel, average FPPs were calculated for each theta-gamma state as reference FPP (as in *Figure 1D*). We then calculated how similar each theta cycle was between the testing and training data by computing the correlation between the testing FPP and the reference FPP. New TG state assignments for individual theta cycles from the testing channel were made and compared to the original assignment to calculate cross-validation accuracy (*Figure 1—figure supplement 5*).

## Pair-wise phase consistency for spike-field analysis

To quantify spike-field phase synchronization, we calculated the pair-wise phase consistency (PPC) between LFP and spikes. PPC is unbiased by the number of trials and less effected by the number of recorded spikes than conventional phase-locking analysis (*Vinck et al., 2012*). In short, PPC for a given frequency band $f$ is calculated with the following equation (for details refer to *Vinck et al., 2012*):

$$PPC(f) = \frac{1}{|M|(|M|-1)} \sum_{m \in M} \sum_{l \in M, \, l \neq m} \frac{\sum_{k=1}^{N_m} \sum_{j=1}^{N_l} U_{k,m}(f) \cdot U_{j,l}(f)}{N_m N_l}$$

in which $|M|$ is number of trials in total, $U_{k,m}$ is the instantaneous phase of filtered LFP at frequency $f$ when the $k$th spike occurs during trial $m$. $N_m$ and $N_l$ are the number of spikes in trial $m$ and $l$, respectively. The instantaneous phase of filtered LFP was calculated through Hilbert transform. PPC was calculated for 6Hz wide frequency bands from 2 to 200 Hz (33 frequency points). Neurons included in the spike field analysis had to satisfy the following two criteria: (1) neurons fired in at least 10 trials during the navigation task and (2) firing rates were higher than 5Hz for interneurons (n=100) and higher than 2Hz for pyramidal cells (n=266), respectively. Putative pyramidal cells and interneurons were classified in the public data set Hc-11 (*Grosmark and Buzsáki, 2016a*) based on their differences in firing rate, peak to trough duration, complex bursting firing and afterhyperpolarization. In total 562 pyramidal cells and 128 interneurons were well sorted and classified.

## Pair-wise phase consistency for LFP-LFP phase synchrony

We calculated the phase synchrony between LFPs recorded in CA1 and EC or CA1 and CA3 on each theta cycle using wavelets in similar way to the spike-field PPC calculation (*Vinck et al., 2012*), as shown previously (*Rohenkohl et al., 2018*). The wavelet cross spectrum $B_k$ between two signals $x(t)$ and $y(t)$ (e.g. signals from CA1 and EC or CA1 and CA3) was calculated for the $k$th theta cycle as $B_k = \{b_{tf} = WC(t,f), \, t = T_k, \, T_k + 1, \, \cdots, T_{k+1}\}$, in which $WC(t,f)$ denotes the wavelet cross spectrum around each local time-frequency point $(t,f)$. Similar to the FPP matrix $A_k(f,\theta)$ defined above, $B_k$ was normalized into a $81 \, (Frequency) \times 20 \, (theta \, phase)$ cross-spectrogram (FPC) matrix, denoted as $B_k(f,\theta)$. Thus $W_k(f,\theta) \triangleq \text{angle}(B_k(f,\theta))$ is the phase lag between the two signals for a given frequency $f$ and theta phase $\theta$. The pair-wise phase consistency for LFP-LFP analysis is then defined as follows:

$$PPC(f,\theta) = \frac{\sum_{k=1}^{N} \sum_{j=1}^{N} W_k(f,\theta) \cdot W_j(f,\theta) - N}{N(N-1)}$$

where $N$ is total number of hippocampal theta cycles. This PPC ranges from -1 to 1 and measures the consistency of the phase lag between the two signals in different frequency bands and theta phases, where -1 denotes no phase locking while 1 denotes a fixed phase lag between two LFPs across all theta cycles.

LFP gamma power is not uniformly distributed across all theta phases and therefore we selected specific phases of theta in which to compute PPC. EC and CA3 are considered source regions of CA1 gamma and, in line with these areas as gamma sources, we observed that the EC-CA1 and CA3-CA1 PPC increased slightly earlier than CA1 gamma power. Thus to measure the possible driving force for CA1 gamma power, we averaged the PPC within a phase interval around the gamma field center but shifted to include earlier phases. Specifically, PPC was averaged from the gravity phase center minus seven phase standard deviations to the gravity phase center plus one phase standard deviation. These phase standard deviations were calculated within the CA1 gamma field.

## Place cell analysis

Firing rate as function of animal position in the environment, or a firing rate map, was computed for putative pyramidal cells. Only running periods were included (speed $\geq$ 5 cm/s). Linear and circular linear tracks were divided into 5 cm bins. Animal position was smoothed through locally-weighted scatter plot smoothing (Lowess) with a 21-sample window (0.525s) (*Hen et al., 2004*). The firing rate map was computed by dividing the spike count map by the occupancy map, after both maps were smoothed with 5-bin boxcar functions. These firing rate maps were used to detect place fields in place cells.

To determine if a cell was a place cell, its spatial information was compared to that of its shuffled spike train. Spatial information of each cell was calculated as:

$$\text{Spatial Information} = \sum_i p_i \frac{\lambda_i}{\lambda} \log_2 \frac{\lambda_i}{\lambda_i}$$

where $\lambda_i$ was the firing rate at the $i$th bin, $\lambda$ was the overall mean firing rate, and $p_i$ was the probability of the animal being in the $i$th bin (occupancy in the $i$th bin / total recording time). Spatial information was calculated based on an adaptive smoothed firing map (as in *Skaggs et al., 1996*). A cell was defined as a place cell when its spatial information was above the 95th percentile of the shuffled data (*Langston et al., 2010*) and its peak firing rate was > 2Hz. To compute shuffled data, for a given cell, spike timestamps were shuffled 100 times by a random interval between 20s and 20s less than the duration of recording session, with the end of the trial wrapped to the beginning to allow for circular displacements.

We performed additional analyses to control for animal speed on neuronal firing rate and spatial information. Animals' behavior was separated into standing still (<5 cm/s), walking (5–15 cm/s), running (15–60 cm/s), and fast running (>60 cm/s) based on the speed distributions of the four theta-gamma states, respectively, specifically based on the observation that S-gamma occurred at a different rate than the other TG states during some speeds (described in Results). Spatial parameters were calculated for speed categories for the four TG states.

Phase precession was calculated within each place field of place cells. A place field was defined as a region consisting at least three adjacent bins with firing rates higher than 20% of the peak firing rate. Place fields were normalized such that 0 denoted the entrance and one denoted the exit of the place field. Phase precession was defined as significant negative linear-circular correlation (p<0.05, linear-circular correlation test) between the animal's position in the place field and the theta phase at the time of the spike (*Berens, 2009*). Linear-circular regression was as in *Kempter et al. (2012)*.

To characterize phase precession in each TG state, we first calculated phase precession using all spikes that occurred within theta cycles that were deemed of that TG state. Next, to control for differences in the number of spikes across different TG states, spikes were randomly deleted from each state until all states had the same number of total spikes. This randomized spike deletion was repeated 100 times to calculate the averaged phase-precession parameter for each place field. Place fields involved in phase precession analysis satisfied the following criteria: (1) significant phase precession; (2) at least 100 spikes occurred in the place field; (3) at least 10 spikes occurred in the place field in any of the TG states. In total 142 out of 223 place fields from 142 place cells were included.

## Acknowledgements

AC Singer acknowledges the Packard Foundation and NIH-NINDS R01 NS109226. CJ Rozell acknowledges NSF grant CCF-1409422. J Lee acknowledges DSO National Laboratories of Singapore. We would like to thank members of the Singer and Rozell labs for feedback on the methods, results, and manuscript. We thank Buzsaki lab members Peter C Petersen, Kenji Mizuseki, and Andres Grosmark for providing information about the Hc-3 and Hc-11 data sets; and Daniel Levenstein, Brendon Watson, and David Tingley for assistance in using their behavior detection tool SleepScoreMaster. We thank Dr. Fabian Koosterman for information about the Hc-19 data set.

# Additional information

## Funding

| Funder | Grant reference number | Author |
|---|---|---|
| National Institutes of Health | R01 NS109226 | Annabelle C Singer |
| David and Lucile Packard Foundation | | Annabelle C Singer |
| National Science Foundation | CCF-1409422 | Christopher Rozell |
| DSO National Laboratories - Singapore | | John Lee |

The funders had no role in study design, data collection and interpretation, or the decision to submit the work for publication.

## Author contributions

Lu Zhang, Conceptualization, Formal analysis, Validation, Investigation, Visualization, Methodology, Writing—original draft, Writing—review and editing; John Lee, Christopher Rozell, Validation, Writing—review and editing; Annabelle C Singer, Supervision, Funding acquisition, Validation, Writing—original draft, Writing—review and editing

## Author ORCIDs

Lu Zhang ⓘD https://orcid.org/0000-0001-7300-1037
Annabelle C Singer ⓘD https://orcid.org/0000-0001-6003-0488

## Ethics

Animal experimentation: All protocols were approved by the Institutional Animal Care and Use Committee of Rutgers University (hc-3) or New York University (hc-11).

## Decision letter and Author response

Decision letter https://doi.org/10.7554/eLife.44320.031
Author response https://doi.org/10.7554/eLife.44320.032

# Additional files

## Supplementary files

• Supplementary file 1. Animal information. Summary of experimental animals involved in the analysis. Note that TG state clustering was performed on all animals from both Hc-11 and Hc-3 data sets. Hc-11 data were specifically used for clustering analysis as well as TG state occurrence and transitions, spike-field analysis and place cell analysis. Hc-3 data were specifically used for CA3-CA1 and EC-CA1 LFP pairwise phase consistency analysis.
DOI: https://doi.org/10.7554/eLife.44320.022
• Transparent reporting form
DOI: https://doi.org/10.7554/eLife.44320.023

## Data availability

All data are available from the CRCNS data repository.

The following previously published datasets were used:

| Author(s) | Year | Dataset title | Dataset URL | Database and Identifier |
|---|---|---|---|---|
| Grosmark AD, Long J, Buzsáki G | 2016 | Recordings from hippocampal area CA1, PRE, during and POST novel spatial learning | https://doi.org/10.6080/K0862DC5 | CRCNS, 10.6080/K0862DC5 |
| Mizuseki K, Sirota A, | 2013 | Multiple single unit recordings from | https://doi.org/10.6080/ | CRCNS, 10.6080/K0 |

| Pastalkova E, Diba K, Buzsáki G | | different rat hippocampal and entorhinal regions while the animals were performing multiple behavioral tasks | K09G5JRZ | 9G5JRZ |
| Ciliberti D, Michon F, Kloosterman F | 2018 | Extracellular recordings from rat hippocampal area CA1 during rest and exploration of a 3-arm maze along with results of real-time and offline detection of hippocampal replay content. | https://doi.org/10.6080/K0PN93T4 | CRCNS, 10.6080/K0PN93T4 |

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
