## [Decision Letter]

Thank you for sending your article entitled "Sub-second dynamics of theta-gamma coupling in Hippocampal CA1" for peer review at *eLife*. Your article is being evaluated by three peer reviewers, and the evaluation is being overseen by a Reviewing Editor and Laura Colgin as the Senior Editor. A consensus was reached based on the original reviews and subsequent discussions between reviewers and editors.

Given the list of essential revisions, the editors and reviewers invite you to respond within the next two weeks with an action plan and timetable for the completion of the additional work. We plan to share your responses with the reviewers and then issue a binding recommendation. The action plan only needs to address the "Essential Revisions" listed below and should contain a similar level of detail as a traditional rebuttal letter.

Summary:

This manuscript investigates cross-frequency coupling of theta and different gamma bands in rodent CA1 using open access data from CRCNS.org. The authors present a novel method for categorizing individual theta cycles and demonstrate that such a categorization can reveal distinct processing states in the CA1 region of the hippocampus. They use a clustering procedure based on the frequency and phase of gamma activity on individual theta cycles, and they identify four distinct theta states (including two "fast gamma" states) during awake behavior and REM sleep. Analyzing transitions between states along with coherence between regions, they find that the network transitions between states depending on behavior and show that subfield coherence occurs during corresponding states. The authors also assess firing rates and spike field coherence of different cell types during different states, finding that slow gamma states have lower firing overall, less spatial information, and less phase precession. Phase precession was most pronounced during medium and "late fast" gamma states.

Examining relationships between rhythms is notoriously hard to characterize at small timescales, and this method could prove useful for determining possible processing state transitions at the hundreds of milliseconds scale.

Essential revisions:

While all the reviewers found the work interesting, several issues arose regarding method details, application of method, and clarity/analyses of their claims.

Specifically, the authors need to address the following:

i) Are the cycle-by-cycle labels reliable/robust/"real"? In other words, can we confidently speak of individual cycles as being in one of the four categories? OR is this just another way of describing theta-gamma coupling that reveals 4 centers in a continuum of possibilities? If this is another way of finding theta-gamma coupling types, the authors need to highlight how their method adds value to the existing literature.

ii) If a potential user of this data applies k-means to data that they've recorded, e.g., with a single tetrode in the pyramidal layer, can they assume with confidence that they will find the same 4 clusters and be able to label theta-cycles? Or do they need the laminar probes?

iii) More analyses need to be done to support some of their claims. In particular, they should verify that clustering is similar during sleep and wakefulness by attempting to identify clusters during those states in isolation (rather than pooling them together). They will need to do this in order to support their claims about the four distinct states in this region.

iv) A concern was raised that the analysis of place cell activity did not adequately control for the location on the track or the animal's speed (subsection “Spiking during S-gamma has lower spatial information and phase precession”, second paragraph) or other potentially important behavioral factors.

v) There is something a little strange about the consistency of LFP and CSD findings across electrodes (and by extension, layers of the hippocampus) as there should be changes across input and cell body layers. Concerns were raised that the authors are potentially not sampling both input and cell body layers or averaging across electrodes that are not from the same location. This needs to be clarified, tightened up, or claims about consistency across layers need to be removed.

vi) Clarification and justification of their clustering, a cleaned up Figure 2, and an exploration of other measures of coherence which are insensitive to power effects are needed.

vii) Particularly given the fact that they've used open data, it would be appropriate for the authors to share their code.

viii) The authors do a good job cataloging how many animals/sessions are used for different comparisons, but it would have been very valuable to really understand how one animal's data differed from another. Authors, for instance, could classify theta cycles using the cluster centers from other animals and compare cluster assignments with what happens when the same animal is used for clustering. Related to this, was any cross-validation done?

[Editors' note: further revisions were requested prior to acceptance, as described below.]

Thank you for resubmitting your work entitled "Sub-second dynamics of theta-gamma coupling in Hippocampal CA1" for further consideration at *eLife*. Your revised article has been favorably evaluated by Laura Colgin as the Senior Editor, a Reviewing Editor, and three reviewers.

The manuscript has been improved but there are some remaining issues that need to be addressed and clarified before acceptance, as detailed below.

Overall, the main concern regards whether the new analyses have been properly cross-validated.

Questions arising regarding novelty and/or robustness of results.

1) “They use community clustering and then switch to k-means clustering.” The reasons for this revolve around robustness to electrode location, but in their revised manuscript, they emphasize the use of the pyramidal layer LFP for analysis. Thus, it seems that the community clustering is superfluous and could be pushed to a comment and/or supplementary figure.

2) “They claim that there are four clusters, but it is unclear to what extent these lie on a continuum or are very distinct.” The authors in response compare model likelihoods for 1 and 4 component models, but they don't carefully describe this in the Materials and methods. Thus, a critical question is whether they have properly cross-validated this result. If they have, this is a valid result. Additionally, they compared inter- and intra-cluster distances, and referred to the "maximum inter-cluster correlation" without defining it. Is it simply the smallest of the largest correlation with any other cluster for that data point? Or somehow for all data points? Furthermore, it is also critical that this analysis be done with cross-validation (i.e., assessing theta-cycles not used to define the model parameters), but this is not specified in the Materials and methods.

3) “They did not assess whether the resulting spectro-temporal descriptions of theta-gamma states were robust across animals.” The authors in response have compared across animals and find that the slow gamma state is quite robust, but the others are not. This reviewer hypothesizes that this may be because the theta oscillation is not consistently phased across different electrodes. If a single theta was used across shanks, would the states be more similar? Additionally, why not just add a tetrode data set from CRCNS to their analysis to ensure that the result is not recording-style dependent?

[Editors' note: further revisions were requested prior to acceptance, as described below.]

Thank you for resubmitting your work entitled "Sub-second dynamics of theta-gamma coupling in Hippocampal CA1" for further consideration at *eLife*. Your revised article has been favorably evaluated by Laura Colgin as the Senior Editor, Frances Skinner as the Reviewing Editor, and additional reviewer Caleb Kemere, who has chosen to reveal his identity.

The manuscript has been improved and will likely be accepted once the remaining issues outlined below are addressed.

1) A question is whether theta phase estimation is less accurate for different gamma bands. If it was, then the quality of the individual cycle estimates might be different, which might affect clustering. The point about medium and fast gamma bands overlapping for certain electrode depths raised this question in a reviewer's mind. If theta power was equivalent across states, then it would suggest that it was equally easy to estimate theta phase, but I suspect that theta power may be different.

Unless the authors have ideas of how to assess the quality of theta phase estimation, the authors are requested to point out in the Discussion that their results depend on accurate theta phase estimates. (I think it may just be the two faster gamma bands.).

The authors should also tone down their claims of four (vs. three) a bit.

Further, the paper emphasizes that it is the first to identify 4 TG states. Authors should at least make sure to also mention that Lopes-dos-Santos et al. reports the potential for 4 TG states and tone down their assertions in this regard.

2) One thing that probably needs to be removed is the 1 vs. 4 Gaussians test. It turns out that for unsupervised learning (which K-means is an example of, but also probabilistic latent-variable models), the log-likelihood, *even with cross-validation*, does not always reflect the best model in model-selection questions. In particular, it will favor models with more components. The reviewer struggled to find a good reference for the authors on this question – both Machine Learning, A Probabilistic Perspective by Murphy, and The Elements of Statistical Learning by Hastie, Tibshirani, and Friedman mention it in passing but don't give much detail. Unfortunately, however, it means that the approach the authors take is not guaranteed to work (particularly, as is described in Murphy, when the underlying data are not actually Gaussian).

3) Several times, the authors assert that they are the first to analyze TG states on a cycle by cycle basis. I believe that Dvorak et al., 2018 do this, as well as Zheng et al., 2016. The authors should definitely at least discuss the Dvorak paper and tone down their assertions in this regard.

4) The authors find that TG states are not affected by novelty. This seems to contradict the findings of Kemere, Carr, Karlsson, and Frank, 2013, and this should be discussed.

5) The authors continue to use the phrase "gravity center". I believe this concept is nearly universally referred to as "center of gravity".

6) In the first paragraph of the subsection “Changes in CA1 PPC with CA3 and EC during different theta-gamma states”, it says "we next coupling". Something is missing.

7) In the second paragraph of the subsection “Spiking during S-gamma has lower spatial information and phase precession”, it says that there was "no spatial preference" of TG states. If there is a speed preference, and a difference in speed across the track (as there must be), how is this possible?

8) The use of Morlet rather than (the better) Morse wavelets may result in suboptimal frequency/phase estimates. The authors should be aware of this.

---

## [Author Response]

[Editors' note: the authors’ plan for revisions was approved and the authors made a formal revised submission.]

Essential revisions:While all the reviewers found the work interesting, several issues arose regarding method details, application of method, and clarity/analyses of their claims.Specifically, the authors need to address the following:i) Are the cycle-by-cycle labels reliable/robust/"real"? In other words, can we confidently speak of individual cycles as being in one of the four categories? OR is this just another way of describing theta-gamma coupling that reveals 4 centers in a continuum of possibilities? If this is another way of finding theta-gamma coupling types, the authors need to highlight how their method adds value to the existing literature.

The reviewers raise an excellent question about whether the underlying theta-gamma (TG) states are four distinct categories or fall along a continuum. Our prior results did not clearly distinguish between whether the different TG states are distinct and each theta cycle clearly falls into one category, or if these TG states are four centers on a continuum and individual theta cycles could fall between these centers. We addressed this question by first determining whether the distribution of frequency phase spectrograms (FPS) for all theta cycles could be better characterized by a 1-Gaussian component model or a 4-Gaussian components mixture model. The former represents the data as one underlying distribution and is consistent with the TG states lying on a continuum (null hypothesis) while the later represents the data as four different distributions and is consistent with theTG states as distinct categories (alternative hypothesis). With this approach we quantified which models fit the sample data better using the likelihood ratio test (LR test). This approach is now described in the Materials and methods section:

“To address whether theta-gamma states are more likely to arise from one distribution or four the likelihood ratio test was applied to a 1-Gaussian component model versus a 4-Gaussian components mixture model:

H0: 𝑝(𝑥) = 𝑁(𝜇, 𝛴)

H1: 𝑝(𝑥) = ∑^4^_𝑘=1_ 𝑤_𝑘_𝑁(𝜇_𝑘_, 𝛴_𝑘_), where 𝑤_𝑘_ > and ∑^4^_𝑘=1_ 𝑤_𝑘_ = 1

𝑁(𝜇, 𝛴) denotes a Gaussian model with mean 𝜇 and covariance matrix 𝛴, where p(x) is the probability density function, and 𝑤_𝑘_ is the Gaussian components weight.[…] As the standard deviation increases, the sample distribution becomes less distinguishable between the 4-Gaussian components and 1-Gaussian model (Figure 1—figure supplement 4).”

We next applied this approach to real data, where the parameters (mean and covariance matrix) of the 4-gaussian models could be directly extracted from the mean and covariance of the samples with each TG states. This is described in the Results section of the manuscript:

“We next wondered whether these four TG states were part of a continuum or were four distinct states. […] Thus our results suggest that the four theta-gamma states are more likely to derive from four distinct underlying distributions than from one distribution.”

We next addressed how different these four TG states are when one considers variation from one theta cycle to the next and this is now described in the Results section:

“We next determined how similar each single theta cycle was to each cluster and if theta cycles could be similar to more than one cluster, as one would expect if theta-gamma states lay on a continuum. […] Overall, our results suggest the four TG states more likely arise from four distinct distributions instead lying along a continuum, while a few of the individual theta cycles may have features of multiple TG states.”

We consider the interpretation of these results in the Discussion:

“We also examined if these theta-gamma states are four centers on a continuum with individual theta cycles could fall between these centers. […] However, this approach is not conclusive as it is not clear how to distinguish between discrete and continuous distributions in real data when those distributions overlap to some extent.”

ii) If a potential user of this data applies k-means to data that they've recorded, e.g., with a single tetrode in the pyramidal layer, can they assume with confidence that they will find the same 4 clusters and be able to label theta-cycles? Or do they need the laminar probes?

Our analysis has consistently shown four theta-gamma clusters from recordings made in the hippocampal pyramidal layer in rat. This data was recorded with 8- or 10-channel Buzsaki probes, which have eight or ten channels clustered at the end of the probe spanning 160 or 200 µm, targeting the hippocampal pyramidal layer. From our analysis such as the results shown in Figure 1—figure supplement 2, we think for recordings in the pyramidal layer (ripples and pyramidal cells are detected) electrode type should not have a significant impact on clustering. Thus we conclude that tetrode recordings aimed at the pyramidal layer would likely produce similar results and laminar probes are not necessary for this analytical approach. We now clarify this in the Results:

“In the data analyzed, the recording sites cover approximately the pyramidal layer (spanning 160 or 200 µm in depth depending number of recording sites in each shank) and do not cover all layers of the CA1 region, like str. rad or str. lm.”

And in the Discussion:

“We conclude that tetrode recordings aimed at the pyramidal layer would likely produce similar results and laminar probes are not necessary for this analytical approach.”

iii) More analyses need to be done to support some of their claims. In particular, they should verify that clustering is similar during sleep and wakefulness by attempting to identify clusters during those states in isolation (rather than pooling them together). They will need to do this in order to support their claims about the four distinct states in this region.

We believe this comment arises from a misconception and we apologize that this was not clear in our manuscript. Awake and REM periods were clustered separately rather than pooling them together. The same clustering procedure was used in each case. We did this because awake and REM theta-gamma coupling could differ, as the reviewers point out and because we have many more samples from awake than from REM periods. In fact, we found similar theta-gamma coupling states in both wake and REM periods however there were slight differences in frequency and theta phase of TG states. We have made this separation of wake and REM periods for clustering clear in the Results section:

“The clustering of TG states described above was done for recordings during awake behavior. We then repeated the same analysis for data recorded from REM periods independently.”

iv) A concern was raised that the analysis of place cell activity did not adequately control for the location on the track or the animal's speed (subsection “Spiking during S-gamma has lower spatial information and phase precession”, second paragraph) or other potentially important behavioral factors.

The reviewers make a good point that the effects of animal speed and location must be considered. We added two supplementary figures to address this question. First, we compared the speed during different TG states and found that S-gamma occurrence is higher than the other states when animals are still or moving very slowly (<5 cm/s) and S-gamma occurrence is lower than other states when animals are running (20-60 cm/s). Based on these differences in TG probability as a function of speed (Figure 5—figure supplement 2A), we separated animals’ speed into 4 types: still (<5 cm/s), walk (5-15 cm/s), run (15-60 cm/s) and fast run (>60 cm/s). We then analyzed neural firing properties for these different speed periods. The firing of neurons is lower in S-gamma states in general than the other TG states across all of these different speeds (Figure 5—figure supplement 2B). Furthermore, the peak firing rate and spatial information of place cells are in general the lowest during S-gamma across all speeds (Figure 5—figure supplement 2CD).

Second, to address whether the animal’s location is playing a role in theta-gamma state analysis, we characterized where different theta-gamma states occur as a function of position in the track. We found the spatial occurrence rates of the four TG states events does not show obvious spatial tuning. We shows two example animals with either low or high spatial information (Figure 5—figure supplement 3). We also computed the spatial information of each TG state. The spatial information ranged from 0.01 to 0.90 bits/theta cycle for all animals which is much lower than that of place cells. We include these findings in the revised Results:

“We also calculated the above parameters when animals traveled at different speeds. We found more S-gamma when animals do not move (Figure 5—figure supplement 2A). Furthermore, animal speed does not seem to account for differences in cells’ firing properties across TG states (Figure 5—figure supplement 2) and we found no spatial preference in the occurrence of TG states events (Figure 5—figure supplement 3).”

v) There is something a little strange about the consistency of LFP and CSD findings across electrodes (and by extension, layers of the hippocampus) as there should be changes across input and cell body layers. Concerns were raised that the authors are potentially not sampling both input and cell body layers or averaging across electrodes that are not from the same location. This needs to be clarified, tightened up, or claims about consistency across layers need to be removed.

First, we agree with the reviewers that these recordings do not sample both input and cell body layers. As we mentioned above, the recording sites in each shank of the probe cover either 160 or 200 µm in depth and are targeted to the pyramidal cell layer. Thus the data is recorded from within the pyramidal layer (with maybe a little area above and below). We now clarify this in the manuscript for example we now describe our results across depths or channels not layers:

“In the data analyzed, the recording sites cover approximately the pyramidal layer (spanning 160 or 200 µm in depth depending number of recording sites in each shank) and do not cover all layers of the CA1 region, like str. rad or str. lm.”

“Four clusters were found in LFPs and theta cycle classification was very similar across different recording depths of the pyramidal layer for each theta cycle (Figure 1—figure supplement 2).”

Second, in revisiting this issue, we found that the consistency in CSD was incorrect: we accidentally normalized the LFPs before we calculated the CSD. We have now corrected this and use raw LFPs to calculate CSD. We now find that there is some variation in CSD across recording depths (Figure 1—figure supplement 3). However, the results of the CSD analysis could be misleading because the recording sites do not span stratum radiatum and stratum lacunosum moleculare, the expected sources of slow and fast gamma, respectively. Thus the CSD analysis results are hard to interpret and we explain this in the manuscript in the Results:

“We also examined current resource density (CSD) across recording depths for the four TG states (Figure 1—figure supplement 3). However because the recording sites do not span str. rad and str. lm, the input layers of CA1 and expected sources of slow and medium gamma, respectively (Colgin et al., 2009), the interpretation of CSD was unclear.”

Third, we also agree that there is variation in recording locations across animals, days, and shanks, while all were targeted at the pyramidal layer. To try to address this variation, we identified the channel with the largest ripple power which should be closest to the pyramidal layer center. From our analysis such as those shown in Figure 1—figure supplement 2, results are consistent across recordings that are in or near the pyramidal layer. Note that in Figure 1—figure supplement 2A (right panel), for recordings deeper than pyramidal center (towards stratum oriens), the four clusters remain and they match the cluster identification from other recording sites based on temporal correlation in Figure 1—figure supplement 2B (right panel). However, the frequency of “medium” gamma (second column in Figure 1—figure supplement 2A, right panel) became higher (as high as high gamma) in recordings from sites towards stratum oriens. Thus we suggest using data recorded near the pyramidal layer center toward stratum radiatum (more superficial) for this analytical approach to preserve the frequency character of “medium gamma”. We also explain this in the Results:

“The results generated by the k-means method with four clusters, were highly robust across all recording sites within the pyramidal layer and were more robust than community clustering (Figure 1—figure supplement 2A, B; right panel). […] Thus we suggest using data recorded near the pyramidal layer center or below (towards stratum radiatum side) for this analysis method to preserve the frequency characteristics of “medium gamma.”

vi) Clarification and justification of their clustering, a cleaned up Figure 2, and an exploration of other measures of coherence which are insensitive to power effects are needed.

Thank you for pointing out these issues. First, we clarified and justified the clustering in the Materials and methods section:

“k-means Clustering: k-means clustering assigns the n power samples 𝐴_𝑖_ in frequency-phase domain to exactly one of k clusters defined by centroids, where k is chosen before clustering (Lloyd, 1982) […] Our Matlab code of the above protocol including other matlab sub functions for clustering individual theta cycles with any LFP signals recorded from hippocampal CA1 region can be found on our lab website (https://singer.gatech.edu/resources/).”

Second, we revised Figure 2 as the reviewer suggested. Specifically, we increased the size of figure elements, eliminated non-significant occurrence and transition results, and color coded the transition probabilities. We believe it is significantly improved.

Third, we clarified that the results of the coherence analysis are not novel, rather the aim of this analysis was to replicate results first reported by Laura Colgin (Colgin et al., 2009) to show that the slow and medium gammas we identified are similar to those originally identified by Colgin (and called slow and fast in that publication). This is now included in the Results:

“The stronger CA3-CA1 synchrony during S-gamma and EC-CA1 synchrony during M-gamma are comparable with previous findings (Colgin et al., 2009), that originally identified slow and medium gammas (called slow and fast gammas in that work).”

Fourth, we used another measure of coherence that is insensitive to power effects and found similar results. Previously, to compute coherence between regions, we used wavelet coherence, which, as the reviewers note, does not clearly distinguish between effects of signal amplitude (“power effects”) and phase on synchrony. To isolate the effects of phase synchrony independent of amplitude, we calculated wavelet phase synchrony instead of wavelet coherence. We implemented pair-wise phase consistency (PPC) methods on LFPs recorded in each subregion (similar to PPC we used for spike-field analysis; see Materials and methods) (Lachaux et al., 2002; LACHAUX et al., 2000; Vinck et al., 2012). Essentially, LFP-LFP PPC quantifies the phase locking between two signals in different frequency bands and is not biased by the number of theta epochs (or duration of analysis periods). These results are updated in Figure 4 as well as in Results section. Both individual examples and the population data are comparable with the previous coherence analysis. Using this PPC method, we observed more obvious CA1-EC synchrony in the medium gamma band during M-gamma states, which is consistent with previous work (Bieri et al., 2014; Colgin et al., 2009; Lasztóczi and Klausberger, 2016; Schomburg et al., 2014).

vii) Particularly given the fact that they've used open data, it would be appropriate for the authors to share their code.

We are happy to share our code. The key algorithms we developed make use of open source code (community detection) and a Mathworks toolbox (k-means) as described in the Materials and methods section. Thus we will share code with examples showing others exactly how to use these open source resources to cluster hippocampal theta cycles (Demo_SingleThetaCluster.m). Author response image 1 is one of the figures generated by our code when applied to one example recording in the hc-11 data set. This code allows others to apply our methods either to data available via CRCNS or to their own data (with tips on how to do in comments in the code). This code is shared via our lab website https://singer.gatech.edu/resources/.

**Author response image 1. respfig1:** FPS from 4 clusters identified using our shared code Solid line: gamma field defined as above 95% of the FPS peak value; triangles: gravity center.

viii) The authors do a good job cataloging how many animals/sessions are used for different comparisons, but it would have been very valuable to really understand how one animal's data differed from another. Authors, for instance, could classify theta cycles using the cluster centers from other animals and compare cluster assignments with what happens when the same animal is used for clustering. Related to this, was any cross-validation done?

This is an excellent point. We now include cross-validation across channels within the same animals and across animals in our revised manuscript. We summarize the results in Figure 1—figure supplement 6, the Results and Discussion sections. In the Results:

“We also performed a cross-validation across channels within the same animals and across animals to understand how clustering differed across recordings. […] The cross validation was in general above chance levels (25% because a given theta cycle could be in 4 states), but highly variable ranging from 0.30 to 0.96 (Figure 1—figure supplement 6 top panel).”

In the Discussion:

“Because we found variability in cross validation analyses across animals and channels, we recommend others apply our method to identified TG states in their own data directly instead of using the TG states we identified as a standard.”

[Editors' note: further revisions were requested prior to acceptance, as described below.]The manuscript has been improved but there are some remaining issues that need to be addressed and clarified before acceptance, as detailed below. Overall, the main concern regards whether the new analyses have been properly cross-validated.We thank the reviewers for their insightful comments. We have addressed these issues and think this has further improved the manuscript. Note that, per the reviewers’ suggestion, we changed Frequency Phase Spectrogram (FPS) to frequency phase power (FPP) and refer to it as such in the responses below.Questions arising regarding novelty and/or robustness of results.1) “They use community clustering and then switch to k-means clustering.” The reasons for this revolve around robustness to electrode location, but in their revised manuscript, they emphasize the use of the pyramidal layer LFP for analysis. Thus, it seems that the community clustering is superfluous and could be pushed to a comment and/or supplementary figure.

Thank you for this suggestion. We have removed almost all discussion of community clustering in the Results and Discussion sections and instead describe details about the use of community clustering in the Materials and methods and one supplementary figure (Figure 1—figure supplement 2).

2) “They claim that there are four clusters, but it is unclear to what extent these lie on a continuum or are very distinct.” The authors in response compare model likelihoods for 1 and 4 component models, but they don't carefully describe this in the Materials and methods. Thus, a critical question is whether they have properly cross-validated this result. If they have, this is a valid result. Additionally, they compared inter- and intra-cluster distances, and referred to the "maximum inter-cluster correlation" without defining it. Is it simply the smallest of the largest correlation with any other cluster for that data point? Or somehow for all data points? Furthermore, it is also critical that this analysis be done with cross-validation (i.e., assessing theta-cycles not used to define the model parameters), but this is not specified in the Materials and methods.

These are excellent points. First, we clarify that maximum inter-cluster correlation is defined as the largest correlation value between a theta cycle’s frequency phase power (FPP) and the mean FPP of every other state (not including the state that the theta cycle is assigned). This is now stated in the Results and Materials and methods sections (see excerpts below).

Second, we now include 5-fold cross-validation of our analysis to determine if the data better fits the 1 or 4 Gaussian components model. To do this, data was separated into training and test data sets. The parameters of the 1 and 4 Gaussian components model were fit with the training data sets (80% of the data), while the likelihood was calculated from the test data sets (20% of the data). In agreement with our prior results, the 4 Gaussian component model better described the test data (p < 10^-9^). We added the following to the Materials and methods:

“To determine if the distribution of FPPs better fit the 1 or 4 Gaussian components models, we performed a 5-fold cross validation. All FPP samples from one given LFP were randomly partitioned into 5 equally sized groups. […] This procedure was repeated five times using each different partition of the data as the test data.”

We also performed cross-validation of the inter- and intra-cluster distances. These results were nearly identical to the previously reported data. We have added following description of the maximum intercluster correlation and cross-validation in the Materials and methods:

“Intra-cluster correlation versus inter-cluster correlation

Intra-cluster correlation was the correlation value between a single theta cycle’s FPP and the mean FPP of the state that theta cycle was assigned. […] Similar results were found with and without cross-validation (data not shown).”

3) “They did not assess whether the resulting spectro-temporal descriptions of theta-gamma states were robust across animals.” The authors in response have compared across animals and find that the slow gamma state is quite robust, but the others are not. This reviewer hypothesizes that this may be because the theta oscillation is not consistently phased across different electrodes. If a single theta was used across shanks, would the states be more similar? Additionally, why not just add a tetrode data set from CRCNS to their analysis to ensure that the result is not recording-style dependent?

Following the reviewers’ suggestion, we calculated the FPP with a single channel in an animal as the reference theta channel and then performed cross-validation within the same animal. This analysis was repeated multiple times with each channel serving as the reference theta for one animal. Finally we pooled all the results together, now reported in Figure 1—figure supplement 5 (shown in first row, light blue and in the third row). We found that using a single fixed theta channel did not improve the overall distribution of cross-validation accuracy. Instead, the cross-validation accuracy became more bimodal with some cases having very high cross-validation accuracy and other cases having very low cross-validation accuracy, near chance levels.

Second, in our previous revision we misunderstood the reviewers’ concerns about whether this method would readily apply to tetrode data. For tetrode recordings there is more uncertainty about the exact recording depth relative to the pyramidal layer than in probe recordings. As previously described our method uses recordings from across the probe to select the channel with the highest ripple power for clustering, and it was unclear how to perform a similar step for selecting an appropriate channel, if any, for tetrode recordings. Therefore, we now test our method on tetrode recordings from the CRCNS hc-19 data set and we report these findings in Figure 1—figure supplement 6 and the Results section:

“We also tested our approach on tetrode data (Figure 1—figure supplement 6). […] Since this step cannot be performed in the same manner for tetrode recordings, we recommend selecting a channel with slow gamma at the peak of theta for clustering analysis when using tetrode recordings.”

[Editors' note: further revisions were requested prior to acceptance, as described below.]The manuscript has been improved and will likely be accepted once the remaining issues outlined below are addressed.1) A question is whether theta phase estimation is less accurate for different gamma bands. If it was, then the quality of the individual cycle estimates might be different, which might affect clustering. The point about medium and fast gamma bands overlapping for certain electrode depths raised this question in a reviewer's mind. If theta power was equivalent across states, then it would suggest that it was equally easy to estimate theta phase, but I suspect that theta power may be different.Unless the authors have ideas of how to assess the quality of theta phase estimation, the authors are requested to point out in the Discussion that their results depend on accurate theta phase estimates. (I think it may just be the two faster gamma bands.).The authors should also tone down their claims of four (vs. three) a bit.

We thank the reviewers for pointing this out. We examined theta power differences across TG-states and did find differences. In particular, lower theta power was found in S-gamma states as Author response image 2 shows. This result is consistent with S-gamma occurring more when animals run slowly. Other states also showed significant differences in power, however it is not clear exactly how such theta power differences contribute to theta phase estimation.

**Author response image 2. respfig2:** Theta Power across TG states. The wavelet spectrum shown (2-20 Hz) was calculated for LFPs from Hc-11 data sets (n=48 channels) during awake pre-maze (left), maze (middle), and post-maze periods (right) over 2-200 Hz. The raw LFP power was normalized by dividing by the total power between 2 Hz and 200 Hz. Colored dots above indicate a TG state was significantly different from all other states (paired t-test, q < 0.05, FDR correction of 186 = 6 states pairs × 31 Frequency sample comparisons).

In the Discussion we now note:

“It is important to note that our method depends on accurate estimation of theta phase. […] Thus, further investigation is needed to establish whether there are three or four distinct theta-gamma states in hippocampal CA1.”

We also toned down the claim of three versus four gammas later in the

Discussion by stating the results are only suggestive:

“Together, these results suggest that EF- and LF-gamma are distinct and that they are two sub-types of fast gamma not previously differentiated.”

Further, the paper emphasizes that it is the first to identify 4 TG states. Authors should at least make sure to also mention that Lopes-dos-Santos et al. reports the potential for 4 TG states and tone down their assertions in this regard.

We previously mentioned Lopes-dos-Santos et al. and have now revised the Discussion to describe this paper and the difference between their results and ours in more detail. Please see our response in 3) for more details and an excerpt from the manuscript.

2) One thing that probably needs to be removed is the 1 vs. 4 Gaussians test. It turns out that for unsupervised learning (which K-means is an example of, but also probabilistic latent-variable models), the log-likelihood, even with cross-validation, does not always reflect the best model in model-selection questions. In particular, it will favor models with more components. The reviewer struggled to find a good reference for the authors on this question – both Machine Learning, A Probabilistic Perspective by Murphy, and The Elements of Statistical Learning by Hastie, Tibshirani, and Friedman mention it in passing but don't give much detail. Unfortunately, however, it means that the approach the authors take is not guaranteed to work (particularly, as is described in Murphy, when the underlying data are not actually Gaussian).

We thank the reviewer for their careful consideration of this point. We removed the log-ratio test of 1 vs. 4 Gaussian models in the revised version as the reviewer suggested.

3) Several times, the authors assert that they are the first to analyze TG states on a cycle by cycle basis. I believe that Dvorak et al., 2018 do this, as well as Zheng et al. 2016. The authors should definitely at least discuss the Dvorak paper and tone down their assertions in this regard.

We now include more discussion of other papers using cycle-by-cycle analysis. We previously cited Lopesdos-Santos et al. and Zheng et al. but now discuss them in more detail with regards to investigating cycle-by-cycle theta oscillations. We also now cite and discuss Dvorak et al. The updated Discussion is as follows.

“Several recent papers have investigated hippocampal theta or theta-gamma coupling on a cycle-bycycle basis (Dvorak et al., 2018; Lopes-dos-Santos et al., 2018; Zheng et al., 2016). […] Thus, our approach could be more suitable for tracking theta-gamma coupling based on both frequency and theta-phase features.”

We also now clarify our use of for the “first time” in the manuscript.

“While others have investigated hippocampal theta or theta-gamma coupling on a cycle-by-cycle basis our method it the first to: (1) classify of theta-gamma coupling using both frequency and phase information and (2) characterize state transition between theta-gamma states.”

4) The authors find that TG states are not affected by novelty. This seems to contradict the findings of Kemere, Carr, Karlsson, and Frank, 2013, and this should be discussed.

Thank you for this suggestion. We have updated the Discussion in the manuscript as follows:

“Previous work revealed that as animals run faster, the frequency of gamma oscillations also tend to be faster (Ahmed and Mehta, 2012; Colgin, 2015; Kemere et al., 2013) […] We observed trends of lower occurrences of S-gamma (p =0.18, t7=-1.47, early vs. middle) and higher occurrences of M-gamma (p=0.08, t7=2.05, early vs. middle) in early trials, when the environment was more novel, although those differences were not significant.”

5) The authors continue to use the phrase "gravity center". I believe this concept is nearly universally referred to as "center of gravity".

We edited the manuscript to use “center of gravity” instead of “gravity center” in the current version.

6) In the first paragraph of the subsection “Changes in CA1 PPC with CA3 and EC during different theta-gamma states”, it says "we next coupling". Something is missing.

We apologize for this error. It has been corrected to state “we next calculated coupling of LFPs”.

7) In the second paragraph of the subsection “Spiking during S-gamma has lower spatial information and phase precession”, it says that there was "no spatial preference" of TG states. If there is a speed preference, and a difference in speed across the track (as there must be), how is this possible?

Thank you for pointing out this issue. We clarified that our claim of “no spatial preference” of TG states is based on the fact that TG states have significantly lower spatial information (<1 bits/theta cycle with 67% of having < 0.1 bits/theta) than place cells (>1.2 bits/spike for more than 90% of the cells). The spatial information and spatial map of TG states was calculated with the same parameters as we used for place cells, namely only including periods when animals ran faster than 5cm/s which is common in the literature (Figure 5—figure supplement 3). As a result differences in TG state occurrence as a function of animal speed were mostly excluded because these differences mainly arose from periods when animals ran at less than 5cm/s (Figure 5—figure supplement 2A). Specifically there were fewer occurrence of Sgamma when animals ran faster than 5cm/s with some additional differences when animals ran at 2060cm/s (Figure 5—figure supplement 2A). If we instead calculate spatial information of TG states including periods when animals ran slower than 5cm/s, we see much higher spatial information (>0.8 bits/theta with 80% >1.2 bits/theta). While these values are now similar to those of place cells they are computed using different types of behaviors. We now clarify this in the Results:

“Additionally, we found low spatial information (<1 bits/theta cycle with 67% of having < 0.1 bits/theta) in the occurrence of TG state events, for periods when animals ran faster than 5cm/s, the same criteria used for calculating the spatial information of place cells (Figure 5—figure supplement 3). […] Furthermore, these results show that S-gamma has lower spatial information and is more likely to occur when animals are moving slowly.”

8) The use of Morlet rather than (the better) Morse wavelets may result in suboptimal frequency/phase estimates. The authors should be aware of this.

Thank you for this reminder. We have added following point in the Materials and methods:

“Note that, although Morlet wavelets are widely applied in LFP analysis, other wavelets such as Morse wavelets might produce better frequency and phase estimates.”